# Nanomolar inhibitor of the galectin-8 N-terminal domain binds via a non-canonical cation-π interaction
Edvin Purić [1], Mujtaba Hassan[2], Fredrik Sjövall [2], Tihomir Tomašič [1], Mojca Pevec [3,4], Jurij Lah [3], Jaume Adrover Forteza[5], Anders Sundin[2], Hakon Leffler [6], Ulf J. Nilsson [2], Derek T. Logan [5] ✉ & Marko Anderluh [1] ✉

Galectin-8 is a tandem-repeat galectin consisting of two distinct carbohydrate recognition domains and is a potential drug target. We have developed a library of galectin-8N inhibitors that exhibit high nanomolar $K_d$ values as determined by a competitive fluorescence polarization assay. A detailed thermodynamic analysis of the binding of D-galactosides to galectin-8N by isothermal titration calorimetry reveals important differences in enthalpic and/or entropic contributions to binding. Contrary to expectations, the binding of 2-*O*-propargyl-D-galactoside was found to strongly increase the binding enthalpy, whereas the binding of 2-*O*-carboxymethylene-D-galactoside was surprisingly less enthalpy-driven. The results of our work suggest that the ethynyl group can successfully replace the carboxylate group when targeting the water-exposed guanidine moiety of a critical arginine residue. This results in only a minor loss of affinity and an adjusted enthalpic contribution to the overall binding due to non-canonical cation-π interactions, as evidenced by the obtained crystal structure of 2-*O*-propargyl-D-galactoside in complex with the N-terminal domain of galectin-8. Such an interaction has neither been identified nor discussed to date in a small-molecule ligand-protein complex.

The noncovalent interactions between small molecules and specific proteins, a key binding mode of most drugs, are still difficult to predict despite extensive research, and there are a number of underestimated noncovalent interactions in protein-ligand complexes[1]. The carbohydrate recognition domain (CRD) of galectin-3 (Gal-3) provides a favourable basic model system for such studies. Even though the affinity of D-galactose (MW 180) by itself for Gal-3 is only about 5 mM, small molecule inhibitors with low nM affinities have been successfully obtained by structure-guided design of synthetic extensions at positions 3 and 1 of D-galactose, yielding an affinity boost of around a million-fold or six orders of magnitude[2]. Along the way, a number of interaction types have been implicated such as hydrogen bonds, cation-π interactions, hydrophobic interactions, halogen bonds and other mechanisms contributing to affinity identified, such as conformational entropy and solvation effects[3].

Preclinical experiments suggest that inhibitors of other galectins would also be of interest, and not redundant. Galectin-8 (Gal-8), the topic here, is a member of the tandem-repeat galectins that contain two nonidentical CRDs, has been shown to modulate both innate and adaptive immune responses[4] and can colocalize with some intracellular pathogens[5]. Since the expression of the Gal-8 gene *LGALS8* is much more amplified in a variety of tumours compared with Gal-1 and Gal-3, possibly its most important function remains its role in cancer[6], particularly in angiogenesis and lymphangiogenesis, and inhibiting these processes by a Gal-8 inhibitor seems a plausible therapeutic option[7].

Gal-8 has two CRDs (C- and N-terminal CRDs) with different affinity and specificity, and inhibitor design has focused on the N-terminal one (Gal-8N) as it has the most distinct features and inhibiting one of the two CRDs should be enough for the inhibition of whole tandem-repeat galectin[8,9]. Furthermore, ligands generally bind to the N-terminal domain (Gal-8N) with substantially higher affinity than to the C-terminal domain (Gal-8C). The present study is part of a program towards nanomolar affinity Gal-8N inhibitors with selectivity over other galectins, thus finding

[1]Department of Pharmaceutical Chemistry, Faculty of Pharmacy, University of Ljubljana, Ljubljana, Slovenia. [2]Department of Chemistry, Lund University, Lund, Sweden. [3]Department for Physical Chemistry, Faculty of Chemistry and Chemical Technology, University of Ljubljana, Ljubljana, Slovenia. [4]Department of Bio-engineering Sciences, Structural Biology Brussels, Vrije Universiteit Brussel, Brussels, Belgium. [5]Department of Chemistry, Biochemistry and Structural Biology, Centre for Molecular Protein Science, Lund University, Lund, Sweden. [6]Department of Laboratory Medicine, Section MIG, Lund University, Lund, Sweden. ✉e-mail: derek.logan@biochemistry.lu.se; marko.anderluh@ffa.uni-lj.si

interaction possibilities most favourable for Gal-8N[10,11]. Like other galectin CRDs, Gal-8N is ~130 amino acid β-sandwich that contains a galectin-defining amino acid motif coordinating conserved binding of galactose via hydrogen bonds to OH 4 and 6, and ring oxygen, and hydrophobic interaction with H 3, 4 and 5[12]. Natural or synthetic extension at position 3 of the galactose will reach into a subsite with most sequence variability among galectin CRDs and, consequently, is the major source of different specificity among galectins and opportunity for selectivity (for detailed information please see the review by Purić et al.[10]). Gal-8N has a unique preference for 3-sialylated Gal-residues in natural glycans here, and a benzimidazole 3-extension provided the best affinity enhancement so far (about 40-fold enhancement in affinity compared to D-galactose), and some selectivity over Gal-3[13]. Extensions at position 1 of the D-galactose will be in a more conserved area of the galectin that typically accommodates D-glucose, N-acetyl-D-glucosamine or N-acetyl-D-galactosamine in natural glycans. Recently, it was discovered that these could be replaced by an α-linked halogenated phenylsulfanyl, which by unique positioning of a halogen bond can boost affinity by 50-200-fold for many galectins[13–15]. Combining extensions at position 3 and 1 of the D-galactose residue afforded the most potent Gal-8N inhibitor (1, Fig. 1) with a $K_d$ value of 1.8 μM (~5000-fold more potent than D-galactose) and, most importantly, a slightly improved selectivity over Gal-3 (2.8-fold) compared to the previously prepared inhibitors[13]. In subsequent work, we searched for a more selective Gal-8N inhibitor, resulting in 3-lactoyl-D-galactoside 2 with a $K_d$ value of 12 μM and a 6.8-fold selectivity over Gal-3[16]. A more recent strategy to mimic D-galactose with D-galactal in compound 3 showed success in improving the selectivity of Gal-8N ligands, with a 14-fold selectivity over Gal-3, but similarly high affinity could not be reached since the affinity boosting addition on position 1 mentioned above was not possible with a galactal[13]. In parallel, Blanchard and co-workers synthesized a series of methyl-β-D-galactomalonic acid derivatives with low micromolar affinities[17,18]. None of these studies have indicated that position 2 of galactose, although pointing into solution, could be of interest for derivatization to increase the affinity and selectivity of Gal-8N.

Therefore, we designed a focused library of D-galactosides by structural modifications of the lead 1 with different substitutions at position 2 of D-galactose of 1 but the same in all other respects. We evaluated the binding of synthesized analogues by fluorescence polarisation assay and isothermal titration calorimetry (ITC) and solved a co-crystallized structure of a selected inhibitor in complex with Gal-8N. This strategy led us to selective nanomolar inhibitors of Gal-8N and a serendipitous discovery that 2-propargyl-D-galactoside binds its target via an unforeseen non-canonical cation-π interaction that does not constitute a classical interaction tool of the medicinal chemist.

## Results
### Design of (selective) galectin-8N inhibitors
In our quest for selective and potent Gal-8N inhibitors, we designed D-galactosides starting from 1, which showed substantial, but not excellent potency and selectivity for Gal-8N over other galectins (Fig. 2)[13]. The crystal structures of methyl 3-O-((7-carboxy)quinolin-2-yl)-methyl)-β-D-galactopyranoside

(PDB ID: 7AEN)[14] and of compound 3 (PDB ID: 7P1M)[14] in complex with Gal-8N show that the 2-hydroxyl group (or the hydrogen in the D-galactal of 3 at the same position) points toward solvent, but with the attachment of an appropriate substituent we aimed to target Trp86, Arg45, Arg59 and/or Arg69 for additional interactions to improve affinity for Gal-8N (Fig. 2). The ligand design involved introducing (i) substituted 2-O-benzyl moieties for potential formation of interactions with Arg45/Arg69 (Fig. 2b), (ii) triazolo-phenyls bearing a carboxylic acid group for potential formation of ionic and/or cation-π interactions with Arg45/Arg59/Arg69 (Fig. 2c), (iii) cycloalkyls for formation of hydrophobic contacts (Fig. 2d), and (iv) a carboxymethylene moiety for formation of a salt bridge with Arg45/Arg69 (Fig. 2e). All designed compounds were docked to the chosen Gal-8N crystal structure (PDB ID: 7AEN)[13] using Glide to help select candidates for synthesis and affinity screening. Eighteen selected compounds (Supplementary Information) were synthesized. The synthesis of all assayed compounds (8a-f, 11, 15a-c, 16a-c, 19, 22, 25, 29, 32) began with a previously reported, methyl ester-protected carboxybenzimidazole 4[13], and is described in detail in the Supplementary information (Supplementary schemes 1–8). All compounds incorporated a 3-O-linked 6-carboxy-benzimidazole to strongly enhance the affinity for Gal-8N[14]. Cycloalkane fragments are popular in drug design where they serve as a core structure or peripheral side chain. They can replace larger cyclic systems, increase metabolic stability, act as aryl isosteres, or simply reduce the entropy and thus increase affinity compared to non-cyclic alkyl counterparts[19]. With this in mind, we hypothesized that 2-O-substitution with cycloalkanes (19, 22, 25) might lead us to derivatives with slightly higher affinities due to van der Waals and hydrophobic interactions. Carboxylates can be involved in interactions with arginine guanidine groups where free binding energy can arise from both ion-pairing and hydrogen bonding, although this might prove to be challenging in water-exposed binding sites, as will be discussed later[3,20,21]. This is why we envisaged a carboxylate-bearing functionality at position 2 of 1 to see whether the water-exposed Arg45 or Arg69 might be efficiently targeted by a tailored carboxylate-bearing 29 (apart from the carboxylate-triazoles 16a-c). The carboxylate-triazoles 16a-c were designed to possess both a phenyl ring and a free carboxylate since a phenyl moiety can offer cation-π interactions while a carboxylate can be involved in ionic interactions or salt bridges with Arg45 and/or Arg69. According to our previous studies[13–15], esters of 6-carboxybenzimidazoles do not possess higher affinities for Gal-8N compared to their carboxylate counterparts and are therefore not appropriate as on-target probes. Rather, we have prepared esters in this study as they may penetrate the cell membrane more rapidly to reach the cytosol, where intracellular esterases are expected to catalyse their hydrolysis and cause ester activation to specifically target intracellular Gal-8[22].

### Affinity evaluation with competitive fluorescence polarisation assay and isothermal titration calorimetry
The results from affinity evaluation of the synthesized carboxybenzimidazoles in a competitive fluorescence polarization assay are presented in Table 1, whereas the affinity evaluation results for their ester counterparts can be found in Supplementary Table 3[8,23,24].

**1**
$K_d$:
**galectin-3**: 5 μM
**galectin-8N**: 1.8 μM
Hassan M. et al., Eur J Med Chem, 2021

**2**
$K_d$:
**galectin-3**: 82 μM
**galectin-8N**: 12 μM
Girardi B. et al., ChemMedChem, 2022

**3**
$K_d$:
**galectin-3**: 690 μM
**galectin-8N**: 48 μM
Hassan M. et al., ACS Med Chem Lett, 2021

**Fig. 1 | Structures of selected galectin-8N inhibitors from our research groups.** Carboxybenzimidazole-D-galactoside 1 showed substantial potency and served as a lead compound, whereas 3-lactoyl-D-galactoside 2 and Carboxybenzimidazole-D-galactal 3 showed improved selectivity compared to the lead 1[13,14,16].

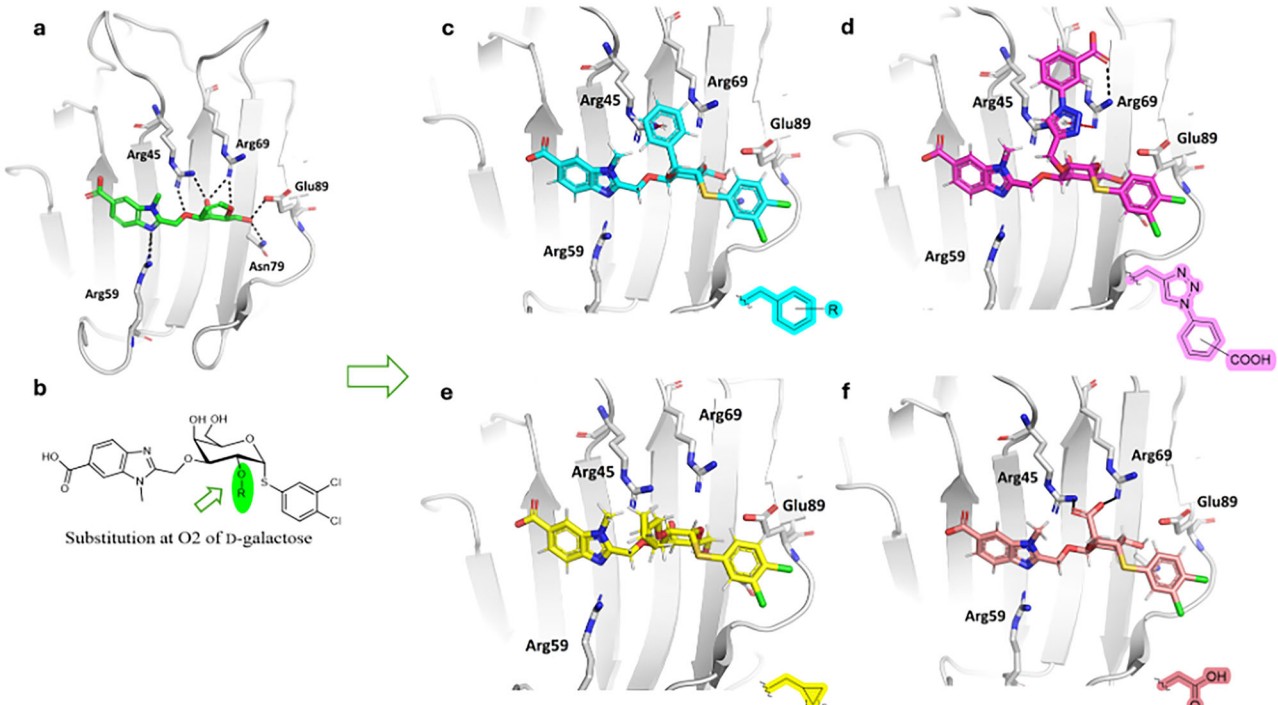

**Fig. 2 | Schematic representation of the design of 2-*O*-substituted galectin-8N inhibitors. a** 3D structure of Gal-8N in complex with D-galactal inhibitor **3** (PDB ID:7P1M; grey cartoon representation of Gal-8N, **3** in sticks: carbon in green, oxygen in red, and nitrogen in blue). For clarity, only residues forming hydrogen bonds (presented as black dashed lines) are shown as grey sticks. **b** Schematic representation of design of new D-galactosides that relies on introduction of new moieties at position 2-*O* of the D-galactoside **1**. **c** Docking binding mode of the designed analogue **8a** with benzyl substitution at 2-OH of the D-galactose. The cation-π interaction between the phenyl ring and the Arg45 side chain is shown as a red dashed line. **d** Docking binding mode of the designed analogue **16b** with triazole-benzoic acid at 2-OH of the D-galactose. The cation-π interaction between the triazole ring and the Arg69 side chain is shown as red dashed line. The hydrogen bond between the carboxylic acid group and Arg69 is shown as a black dashed line. **e** Docking binding mode of the designed analogue **22** with cycloalkyl substituent at 2-OH of the D-galactose. **f** Docking binding mode of the designed analogue **29** with methylenecarboxylic acid at 2-OH of the D-galactose. Interactions with Arg45 and Arg69 are presented as black dashed lines.

The affinity evaluation of compound **11** yielded a very intriguing result. This compound was originally synthesized as a precursor towards triazoles **15a-c** and **16a-c**. However, it turned out that a simple propargyl-bearing **11** had approximately the same affinity as the triazoles **16a-c** and the carboxylate-bearing **29**, while it exhibited much better ligand efficiency (LE) and ligand-lipophilicity efficiency (LLE) compared to the triazoles **16a-c**, as shown in Supplementary Table 5. An interaction comparison with the triazoles **16a-c** is not straightforward due to large structural differences. Therefore, a direct comparison of $K_d$ and thermodynamic parameters between the 2-*O*-propargyl derivative **11**, the 2-*O*-unsubstituted **1** and the 2-*O*-carboxymethylene **29** was performed using isothermal titration calorimetry (ITC) (Fig. 3). The affinities of **1**, **11** and **29** determined by ITC (Fig. 3c) agree quite well ( <3-fold difference) with the affinities obtained by fluorescence polarization measurements, confirming the fluorescence polarization assay (Table 1) as a reliable screening test. Thermodynamic analysis clearly shows that the free carboxylate of **29** affects binding by increasing the enthalpic contribution to the free binding energy, with a slight loss of entropy compared to **1**. Unexpectedly, the propargyl group contributes to an even more enthalpy-driven binding of **11**, with a higher entropic penalty upon binding compared to **29**. These observations are not consistent with previous observations that affinity enhancement by ionic interactions with solvent-exposed arginine side chains is entropy-driven, since the change in binding entropy here is unfavourable[3,21]. This also depends on how well-ordered arginines in the unbound state are.

## X-ray crystallography, quantum mechanical calculations and energy decomposition analysis

To rationalize the unexpected enthalpy-induced increase in affinity due to the propargyl group, the binding mode of **11** was analysed by X-ray crystallography in complex with the Gal-8 N-terminal domain (Fig. 4a, b), followed by quantum mechanical calculations on the same crystal structure with Jaguar (Schrödinger suite, Fig. 4c).

The crystal structure of **11** in complex with Gal-8N was solved to 1.08 Å resolution (PDB ID: 9FYJ) and the compound is bound with full occupancy to both copies of Gal-8N in the asymmetric unit, which allows a detailed analysis of the binding pose (Fig. 4a). This reveals essentially the same interaction pattern as seen in previous crystal structures (Fig. 2a): The 3,4-dichlorophenyl aglycon forms a halogen bond with the Gly87 main chain carbonyl group, D-galactose makes a hydrogen bond network with Arg45, Arg69 and His65, while 3-*O*-carboxybenzimidazole fits the sub-pocket delineated by Asp49, Gln47, and Arg59, with the carboxylate forming a crystal water-bridged hydrogen bond network with several water molecules and amino acid residues involved (Fig. 4b)[14]. The alkynyl moiety of the 2-*O*-propargyl moiety (**11**) leans directly towards the Arg45 guanidinium group with a distance between the terminal propargyl carbon atom and the terminal Arg45 nitrogen atom of 3.2 Å and 3.4 Å in the two independent copies (Fig. 4c). In order to test the hypothesis that the propargyl interacts with guanidinium side chain via electrostatic interaction and not a partial covalent bond with orbital overlap, quantum mechanical calculations on the crystal structure (the ligand was left intact) were performed using Jaguar (Schrödinger suite). The molecular orbitals of **11** and Arg45 were calculated at the B3LYP-D3_6-31G** level. The calculated HOMO/LUMO orbitals are presented in Fig. 4c. and supported the proposed interaction. Furthermore, to dissect specific binding energy contributions of compound **11** in complex with Gal-8N, a quantum mechanics Energy Decomposition Analysis (EDA) was performed[25]. Such an analysis requires that the structure is split into two interacting fragments. Since the compound **11**

**Table 1 | $K_d$ values (µM)[a,b] of 1, 8a-f, 11, 15a-c, 16a-c, 19, 22, 25, 28, 29, and 32**

| Compound | R₂ ($K_d$ values (µM)) | Gal-8N | Gal-3 (selectivity ratio[c]) | Gal-1 (selectivity ratio[c]) | Gal-8C (selectivity ratio[c]) |
|---|---|---|---|---|---|
| 1[13] | -H | 1.8 ± 0.10 | 5.0 ± 1.1 (2.8) | 130 ± 11 (72) | 760 ± 160 (422) |
| 8a |  | 1.5 ± 0.13 | 3.7 ± 0.50 (2.5) | 14 ± 0.82 (9.4) | 320 ± 12 (213) |
| 8b |  | 2.1 ± 0.10 | 8.2 ± 0.70 (3.9) | 19 ± 0.67 (9) | 303 ± 19 (144) |
| 8c |  | 2.1 ± 0.15 | 5.9 ± 0.60 (2.8) | 16 ± 0.75 (7.6) | 256 ± 24 (122) |
| 8d |  | 1.6 ± 0.060 | 4.2 ± 0.60 (2.6) | 14 ± 1.2 (8.8) | 301 ± 28 (188) |
| 8e |  | 4.3 ± 0.40 | 12 ± 3.0 (2.8) | 20 ± 0.84 (4.7) | 577 ± 4 (134) |
| 8f |  | 1.8 ± 0.18 | 5.8 ± 0.70 (3.2) | 15 ± 1.3 (8.3) | 225 ± 37 (125) |
| 11 |  | 0.80 ± 0.090 | 1.4 ± 0.20 (1.75) | 10 ± 0.71 (12.5) | 390 ± 14 (488) |
| 15a |  | 3.5 ± 0.37 | 4.7 ± 1.0 (1.3) | 69 ± 9.5 (19.7) | 355 ± 57 (101) |
| 15b |  | 4.5 ± 0.24 | 22 ± 5.5 (4.9) | 104 ± 7.4 (23) | 568 ± 23 (126) |
| 15c |  | 4.1 ± 0.33 | 23 ± 4.9 (5.6) | 99 ± 3.1 (24) | 350 ± 60 (85) |

**Table 1 (continued) | $K_d$ values (µM)[a,b] of 1, 8a-f, 11, 15a-c, 16a-c, 19, 22, 25, 28, 29, and 32**

| Compound | R2 | Gal-8N | Gal-3 (selectivity ratio[c]) | Gal-1 (selectivity ratio[c]) | Gal-8C (selectivity ratio[c]) |
|---|---|---|---|---|---|
| | | $K_d$ values (µM) | | | |
| 16a | | $0.46 \pm 0.030$ | $1.2 \pm 0.20$ (2.6) | $26 \pm 1.1$ (57) | $159 \pm 4$ (346) |
| 16b | | $0.65 \pm 0.050$ | $2.9 \pm 0.40$ (4.5) | $37 \pm 1.1$ (57) | $100 \pm 2$ (154) |
| 16c | | $1.1 \pm 0.070$ | $6.2 \pm 1.1$ (5.6) | $45 \pm 0.80$ (41) | $201 \pm 37$ (183) |
| 19 | | $4.1 \pm 0.61$ | $21 \pm 1.5$ (5.1) | $76 \pm 15$ (18.5) | $662 \pm 27$ (161) |
| 22 | | $1.0 \pm 0.060$ | $5.7 \pm 1.3$ (5.7) | $12 \pm 1.0$ (12) | $374 \pm 38$ (374) |
| 25 | | $1.6 \pm 0.070$ | $9.4 \pm 1.9$ (5.9) | $16 \pm 0.50$ (10) | $383 \pm 21$ (239) |
| 29 | | $0.50 \pm 0.030$ | $3.3 \pm 0.20$ (6.6) | $49 \pm 3.6$ (98) | $449 \pm 39$ (898) |
| 32 | -CH3 | $1.3 \pm 0.050$ | $4.5 \pm 0.70$ (3.5) | $16 \pm 0.80$ (12.3) | $358 \pm 29$ (275) |

[a]Results represent the mean ± SEM of n = 4 to 8.
[b]$K_d$ was determined by competitive fluorescence polarisation assay.
[c]Ratio between affinity for Gal-8N vs. galectin of choice: $K_d$ (Gal-X)/$K_d$ (Gal-8N).

has multiple interactions with the protein and we wanted to investigate the interaction between Arg45 and the acetylene, the ligand had to be modified. The first fragment was chosen to be the galactose C2 substituent bearing the acetylene, and second fragment was Arg45 together with any parts of the protein or ligand that may influence the electronic structure of Arg45 (Fig. 5a). That is, the stacking Arg69 with its water-mediated salt bridge to Glu89 and the hydrogen bond network from Arg45 through galactose 4-OH to His65 to Asp49 were included. The Molecular Orbital Energy-Level Diagram obtained by the EDA analysis

(Fig. 5b) indicates that the acetylene and Arg45 fragments have a π-π* interaction and the predicted bonding energy is −3.4 kcal mol⁻¹ (please see Supplementary Data 1). Plotting the bonding molecular orbital indicates a π–π* interaction between the acetylene and the guanidinium ion (Supplementary Fig. 109). It has previously been shown that the LUMO of the guanidinium ion is π*[26]. The observation was corroborated by a Natural Orbitals for Chemical Valence (NOCV) deformation density plot that describes the charge flow between acetylene and Arg45 when the two fragments are combined. Electrons flow from red to blue,

**Fig. 3 | ITC analysis. a** Calorimetric titration curves measured at 25 °C by injections of compounds **1** (black squares)**, 11** (red), and **29** (blue) into the Gal-8N solution. Number of replicate titrations: $n = 2$. **b** Bar chart and **c** the table with standard thermodynamic parameters of **1**, **11** and **29** binding to Gal-8N determined by fitting of 1:1 binding model to the titration curves presented in (**a**). ± Values are standard deviation obtained by Monte Carlo error analysis.

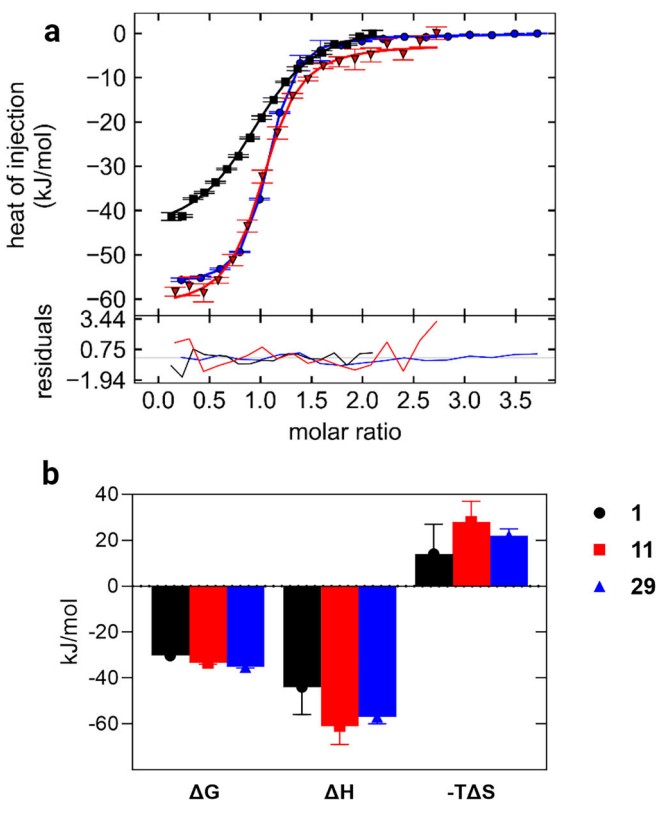

| Compound | $\Delta G$ (kJ/mol) | $\Delta H$ (kJ/mol) | $-T\Delta S$ (kJ/mol) | $K_d$ (µM) |
|---|---|---|---|---|
| **1** | −30.2 ± 0.6 | −44 ± 12 | 14 ± 13 | 5 ± 1 |
| **11** | −33.5 ± 0.5 | −61 ± 8 | 28 ± 9 | 1.3 ± 0.3 |
| **29** | −35.3 ± 0.4 | −57 ± 3 | 22 ± 3 | 0.6 ± 0.1 |

i.e. from acetylene towards the guanidinium group, thus showing the origin of π-π* interaction (Fig. 5c). A specific dissection of energy bonding contributions is given as a separate supplementary file (Supplementary Data 1 - Puric et al. - Bonding Energy).

## Discussion

Our quest for potent Gal-8N inhibitors began with the design of a focused compound library starting from **1**. Although we are aware that scoring functions rarely predict the correct binding free energy, we envisaged that the docking protocol should be able to accurately predict the isolated influence of substituent variations at the 2-O-position of the D-galactose core. With this in mind, we focused on the synthesis of a limited number of derivatives of **1** with rather simple 2-O-substitutions. The resulting compounds **8a-f**, **10**, **11**, **15a-c**, **16a-c**, **19**, **22**, **25**, **26**, **29**, and **32** were evaluated for their binding affinity using an established competitive fluorescence protein-binding assay. The results in Table 1 indicate that the 2-O-benzyl-bearing compounds **8a-f** and the 2-O-triazole-bearing **15a-c** did not increase the binding affinity compared to our previous hit **1**[13]. Of the 2-O-(cyclo)alkyl derivatives, the 2-O-cyclopropyl derivative **22** appears to be the best fit to Gal-8N with a $K_d$ value of 1.02 µM. This is consistent with the very flat and water-exposed binding site of Gal-8N, where substituents forming van der Waals and hydrophobic interactions are expected to provide a modest improvement in binding. Carboxylates **16a-c** showed a higher affinity for Gal-8N, particularly **16a** with a $K_d$ value of 460 nM. This is a reasonable improvement over lead **1** and can be explained by the expected salt bridges between the carboxylates of **16a-c** and the guanidine moieties of the Arg45 or Arg69 side chains (Fig. 2c). Virtually the same affinity was

measured for the 2-O-carboxymethylene derivative **29** ($K_d$ value of 500 nM; ITC measurements showed an order of magnitude improvement over **1**), most likely due to the same type of interactions (Fig. 2e), and these molecules are the very first Gal-8N inhibitors with nanomolar $K_d$ values and reasonable, but not excellent, selectivity over Gal-3 (up to 6.6-fold). Despite these improvements, we must emphasize that the selectivity over other galectins in the entire series was improved by a maximum factor of 2. This indicates that modifications of positions 1 and 3 of the D-galactose core are still the best option for the development of galectin inhibitors with significantly improved affinity and selectivity, and the overall affinity and selectivity is the consequence of a finely tuned combination of substitutions at positions 1, 2 and 3.

Compound **11** showed some selectivity for Gal-8N over Gal-1 and -3 with about 13-fold and 2-fold selectivity, respectively. However, the most intriguing feature of **11** is its affinity for Gal-8N, as it reached high nanomolar affinity for Gal-8N ($K_d$ value of 800 nM), which is somewhat unexpected when comparing **11** with **29**. Indeed, thermodynamic analysis (Fig. 3b and c) shows that the 2-O-carboxymethylene moiety (with a free carboxylate) of **29** affects binding with a substantial increase in binding enthalpy and a slight loss of entropy compared to **1** ($\Delta\Delta H = -13$ kJ mol$^{-1}$, $T\Delta\Delta S = +8$ kJ mol$^{-1}$). This was not expected, as we found in our previous work that water-exposed arginines form entropy-driven ionic (Coulomb) interactions, yet a direct comparison with the data presented in this work is not trivial since in the previous work the ionic interaction involved the guanidine and sulphate groups and the latter (sulphate groups) are not present in our current set of molecules[3]. The reason for this could lie in the specificity of the Gal-8N binding site, which is rather flat and hydrated with

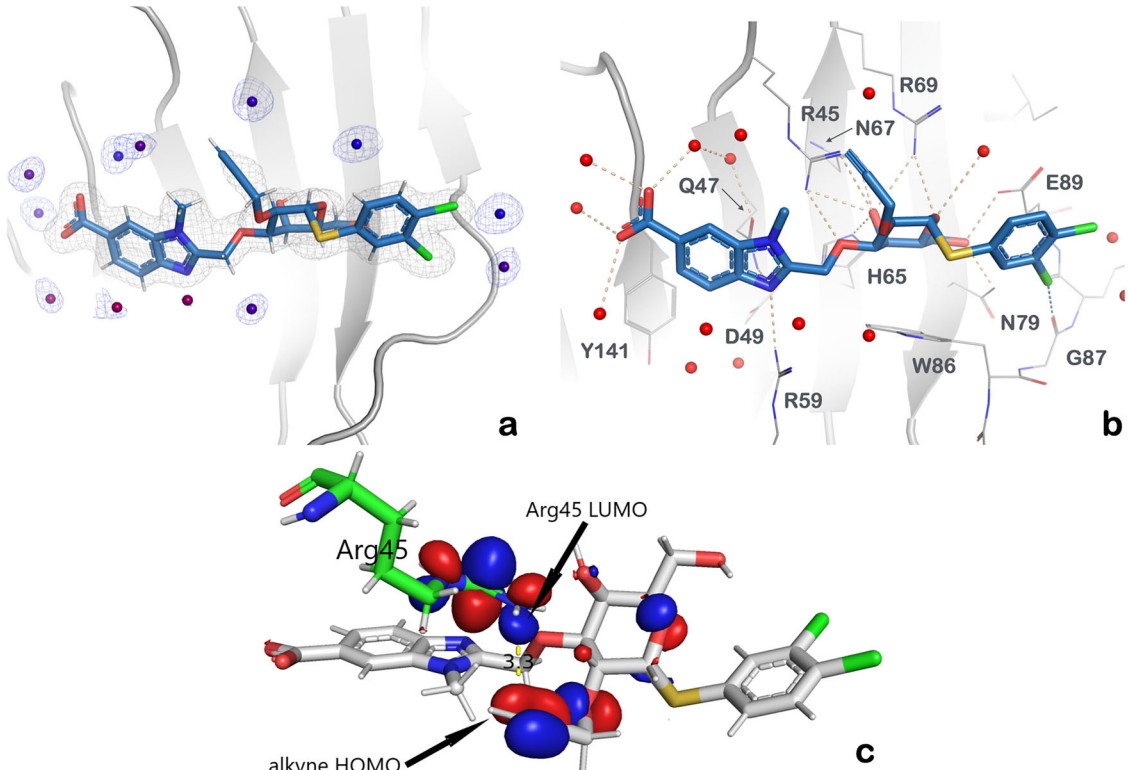

**Fig. 4 | Electron density map, crystal structure and quantum mechanical calculations of 11 in complex with galectin-8N. a** 2m|Fo|–D|Fc| electron density map. **b** Crystal structure of **11** in complex with Gal-8N (PDB ID: 9FYJ): green cartoon representation of Gal-8N, **11** in sticks: carbon in light blue, oxygen in red, and nitrogen in blue with amino acid residues that from binding pocket exposed (in sticks: carbon in grey, oxygen in red, and nitrogen in blue). **c** Quantum mechanical calculations on the same crystal structure of the Gal-8N–**11** complex using Jaguar (Schrödinger suite). The calculations revealed that the LUMO of Arg45 of Gal-8N (depicted in blue) does not overlap with the HOMO of the alkyne from **11** (depicted in red) indicating that the true origin of the interaction is most probably electrostatic.

loosely bound water molecules with high internal energy (and therefore not stable in their environment), termed "hot" water molecules. The displacement of these loosely bound water molecules by a ligand into the bulk phase with fully tetra-coordinated state would lead to enthalpy gain and entropic penalty[27]. The increase in entropy is contrary to the prevailing medicinal chemistry strategy in ligand design, where the best-in-class drugs/ligands should be designed to bind their targets in an enthalpy-driven manner, as reported by Freire[28]. Surprisingly, the comparison of thermodynamic parameters between **11** and **1** (Fig. 3b, c) shows an unexpected gain in enthalpic contribution with a further loss in binding entropy compared to **29** (**11** vs **1**: $\Delta\Delta H = -17$ kJ mol$^{-1}$, T$\Delta\Delta S = +14$ kJ mol$^{-1}$, **11** vs **29**: $\Delta\Delta H = -4$ kJ mol$^{-1}$, T$\Delta\Delta S = +6$ kJ mol$^{-1}$), suggesting that contributions other than classical hydrophobic interactions govern the binding of **11**. The important question to be answered is what is the origin of the observed enthalpic gain, especially when comparing **11** with **29**? MD simulations of **11** and **29** bound to Gal-8N showed that the complexes of Gal-8N with **11** and **29** have approximately the same stability (Fig. S98–S103) and that the 2-*O*-propargyl moiety (**11**) is slightly more flexible in the bound state than the 2-*O*-carboxymethylene (**29**) (Fig. S100 and Fig. S103). The latter could explain why **29** has a lower entropic penalty than **11** upon binding, but it could also be the result of steric hindrance of the substituent at position 2, which could affect the flexibility of the substituents at positions 1 and 3. To exclude this possibility, we performed the MD simulation of **1**, **11** and **29** in water only (Fig. S104–S106). Comparison of the MD simulation data of **1** with **11** and **29** shows that the bond torsional profiles of the substituents at positions 1 and 3 remain almost the same regardless of the substituent at position 2, indicating that the substituents at position 2 of **11** and **29** leave the flexibility of the substituents at positions 1 and 3 almost unaffected. However, an even more important answer that could explain the increase in binding enthalpy comes from the structural analysis of **11** in complex with Gal-8N (Fig. 4a), where a contact between the 2-*O*-propargyl moiety and a

guanidine moiety of Arg45 is observed (Fig. 4c). This contact is similar to, but not the same as a contact we observed between D-galactal ethylene and the guanidine of Arg45 and may be rationalized by non-canonical cation-π interactions. Alkyne-cation interactions have been reported several times in synthetic chemistry[29,30], but to our knowledge no such interactions have been found between a ligand and a protein, which is why we have labelled them "non-canonical cation-π interactions". These were studied with quantum mechanical calculations which indicate that the LUMO of Arg45 of Gal-8N (Fig. 4c, depicted in blue) points towards the HOMO electrons of propargyl alkyne (Fig. 4c, depicted in red) of compound **11**, but without evident HOMO-LUMO overlap, meaning that the covalent nature of this interaction is negligible. Yet, the distance between the guanidine C-atom and the terminal C-atom of the alkyne is 3.3 Å (Fig. 4c), which is in line with a distance threshold for classical cation-π interactions, as these occur within a distance of 6 Å[31]. The observed interaction is unique and may account for the enthalpy-driven increase in the affinity of **11** for Gal-8N. The reason why **11** binds with a higher enthalpy than **29** is counterintuitive but can be explained by the fact that the ethynyl moiety has a lower desolvation enthalpy than the free carboxylate (where release of tightly coordinated waters around carboxylate can lead to a loss in enthalpy and gain in entropy), resulting in a higher overall binding enthalpy. A unique property of the ethynyl moiety π-system was acknowledged by Wilcken et al. who described its halogen-like properties and suggested that the ethynyl should be considered as a halogen bioisostere because the partially positively charged terminal ethynyl hydrogen resembles a positively charged halogen σ-hole[32]. Unfortunately, their switch from iodine to ethynyl substituent led to a significant loss of affinity for p53 cancer mutant Y220C (17.5-fold). On the contrary, our work shows that the ethynyl group can successfully replace the carboxylate in targeting the water-exposed guanidine group with a negligible loss of affinity, but with tuned enthalpic contribution to overall binding energy due to specific cation-π interactions. These unforeseen non-

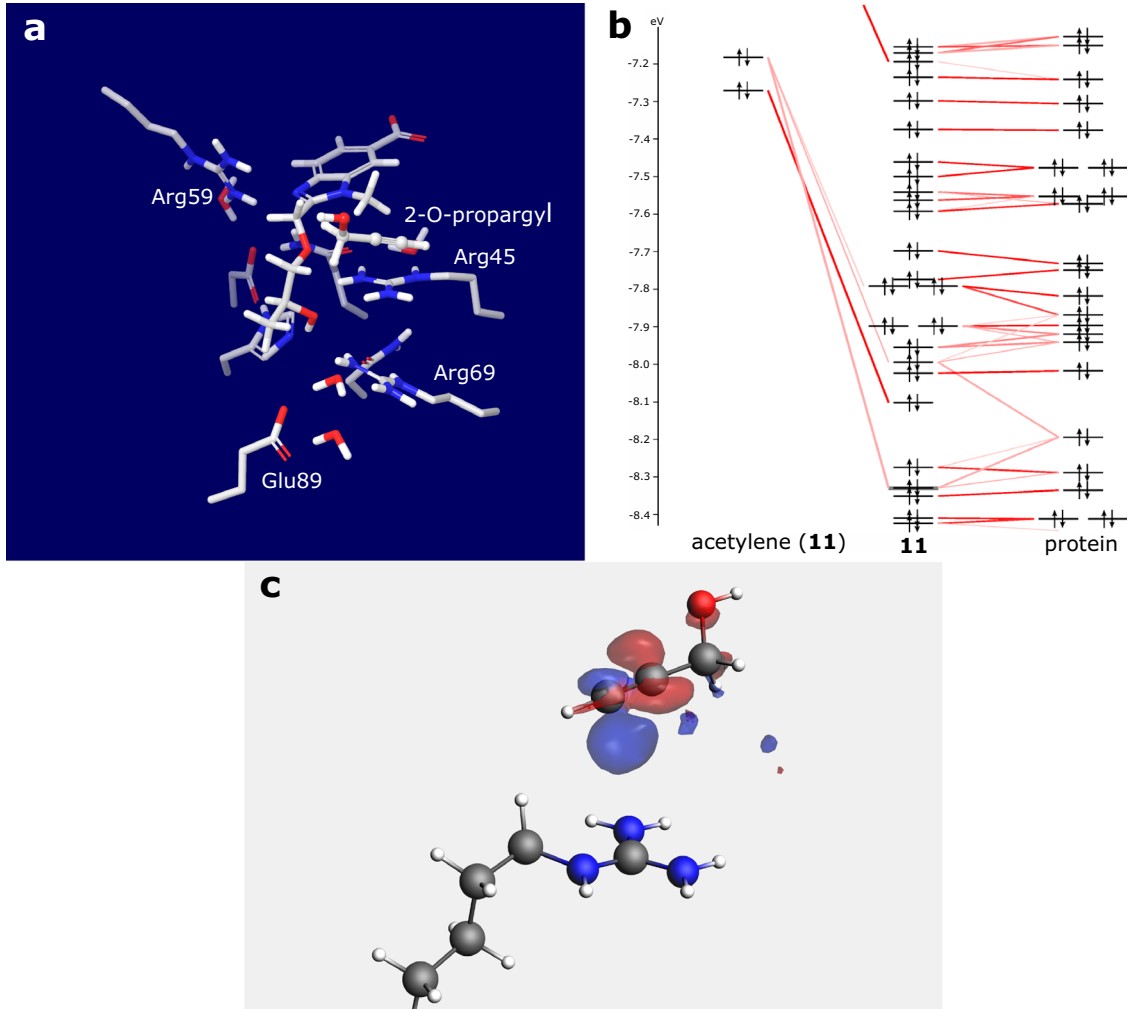

**Fig. 5 | Energy Decomposition Analysis (EDA). a** The two fragments chosen for EDA analysis. The first fragment is depicted as ball and stick, the second fragment as stick bonds. **b** The Molecular Orbital Energy-Level Diagram obtained by the EDA analysis indicates that the acetylene and Arg45 fragments have a bonding interaction as the bonding interaction between the two fragments is generated by a new molecular orbital. **c** A natural orbitals for chemical valence (NOCV) deformation density plot describes the charge flow between acetylene and Arg45 when the two fragments are combined. Electrons flow from red to blue, i.e. from acetylene towards the guanidinium group.

canonical cation-π interactions are not a part of the current medicinal chemist interaction tools. Our results have important general implications for ligand design, as the correct choice of arginine interacting partners influences the thermodynamic footprint of the ligands by switching from entropy- to enthalpy-driven interactions and offers a specific guideline that can be systematically exploited in drug design.

## Methods
### Molecular docking calculations
Molecular docking calculations were performed using Schrödinger Release 2022-1 (Schrödinger, LLC, New York, NY, USA, 2022). The co-crystal structure of Gal-8N in complex with inhibitor **1** (PDB entry: 7P1M) was prepared using Protein Preparation Wizard with the default settings: bond orders were assigned using the CCD database, missing hydrogens were added, termini were capped, the missing side chains were modelled with Prime, and the protonation states (pH 7.0 ± 2.0) were modelled with Epik[33]. The receptor grids were calculated for the ligand-binding site. Ligand structures were prepared using LigPrep module and ionized with Epik at pH = 7.4 using OPLS4 force field. The compounds were then docked using the Glide XP protocol as implemented in Schrödinger Release 2022-1 (Glide, Schrödinger, LLC, New York, NY, USA, 2022). The highest scored docking conformation was used for analysis and presentation.

### Molecular dynamics simulations
Gal-8N in docking complex with inhibitor **29** or crystal structure of **11** in complex with Gal-8N (PDB entry: 9FYJ) was used as an input for molecular dynamics simulation using Desmond[34]. The systems for simulations of the protein-ligand complexes or compounds **1**, **11** and **29** were prepared with System Builder. TIP4P water molecules up to 10 Å from the protein surface or ligand were added to solvate the system in an orthorhombic box. Solvated system was then neutralized by adding sodium and chloride ions at a concentration of 0.15 M. OPLS_2005 force field was used for parametrization of the protein-ligand complex[35]. The default Desmond relaxation protocol was used for the equilibration stage: (1) 100 ps of Brownian dynamics NVT, 10 K, small timesteps, with restraints on the solute heavy atoms, (2) 12 ps NVT, 10 K, with small timesteps and restraints on the solute heavy atoms, (3) 12 ps NPT, 10 K, and restraints on the solute heavy atoms, (4) 24 ps unrestrained NPT. The equilibration was followed by the 300 ns long production stage (done in two replicates): NPT ensemble at 300 K and 1.013 bar pressure with Langevin thermostat and barostat (1 and 2 ps relaxation time, respectively), RESPA integrator with 2 fs time step, cut-off scheme at 9.0 Å. Molecular dynamics trajectories were analyzed using the Simulation Interactions Diagram algorithm in Maestro. Affinity clustering was performed according to the work by Frey and Dueck[36].

## Quantum mechanical calculations

In order to investigate the nature of interaction between the propargyl moiety of **11** and Arg45 side chain, a QM calculation was performed on the crystal structure of Gal-8N in complex with **11**. The protein was simplified to residue Arg45. A single point energy calculation at the B3LYP-D3_6-31G** level was performed and the molecular orbitals from HOMO -30 to LUMO+30 were calculated with Jaguar implemented in the Schrödinger release 2022-1. Plotting HOMO-21 and LUMO+1 indicated the presence of a non-canonical interaction between Arg45 side chain and the alkyne bond of the propargyl moiety.

## Energy Decomposition Analysis

To explore the interaction between Arg45 and the acetylene of compound **11**, an Energy Decomposition Analysis was performed between two fragments as depicted in Fig. 5a. The Amsterdam Modeling Suite (AMS 2024.1, SCM, Amsterdam, The Netherlands, http://www.scm.com) was utilized for this purpose. All QM single point energy calculations were performed with the BP86 functional with D3(BJ) dispersion correction and using the TZ2P triple-zeta basis set. ZORA scalar relativistic effects were included in the calculations and the numerical quality was set to Very Good. Bonding Energy report is attached as a separate file (Supplementary Data 1 - Puric et al. - Bonding Energy).

## Chemistry

Commercially available reagents (Apollo Scientific, TCI, Sigma-Aldrich, Acros Organics, Angene Chemistry, Enamine, BLD Pharm and Fluorochem Ltd.) were used without any further purification. Thin-layer chromatography (TLC) analysis was performed on Merck 60 F254 silica gel plates (0.25 mm) under visualization with UV light and phosphomolybdic acid (PMA) stain. Flash column chromatography was performed on silica gel 60 with particle size 240–400 mesh. Compounds were detected at 254 nm wavelength. $^1$H, $^{13}$C, and $^{19}$F NMR spectra were recorded at 400 MHz, 101 MHz, and 376 MHz, respectively, using a Bruker AVANCE III 400 MHz NMR spectrometer (Bruker Corporation, Billerica, MA, USA) at ambient temperature in CDCl$_3$, Acetone-$d_6$, DMSO-$d_6$, Acetonitrile-$d_3$, and D$_2$O with tetramethylsilane as an internal standard. Chemical shifts are reported in δ parts per million (ppm), with multiplicity, coupling constants (in Hz) and integration. High-resolution mass spectrometry (HRMS) was performed using Exactive Plus Orbitrap, Thermo Fisher Scientific, Waltham, MA, USA. Reversed-phase high-performance liquid chromatography (HPLC) analysis was performed on Thermo Scientific Dionex UltiMate 3000 modular system (Thermo Fisher Scientific Inc., MA, USA), where a C18 column (1.8 μm, 2.1 mm × 50 mm; UPLC® HSS C18 SB column) was used, with a flow rate of 0.4 mL min$^{-1}$ and sample volume of injection of 1–5 μL (mobile phase: 0.1% trifluoroacetic acid (TFA) [v/v] in ultrapure water (solvent A) and acetonitrile (CH$_3$CN) (solvent B) with a gradient of 10-90% B, $T = 40$ °C). Alternatively, the purities of compounds were monitored by liquid chromatography-mass spectrometry, which was performed using method A. The system was coupled to the mass spectrometry (Expression CMSL; Advion Inc., Ithaca, NY, USA). All tested compounds were ≥ 95% pure (according to the HPLC purity).

Method A: A C18 column (Waters xBridge BEH; 4.6 mm × 150 mm, 3.5 μm) was used at 40 °C. The flow rate of mobile phase was 1.5 mL min$^{-1}$ (injection volume: 10 μL) and the products were detected at 254 nm. Solvent A: 1% CH$_3$CN and 0.1% HCOOH in double-distilled H$_2$O; Solvent B: CH$_3$CN. The following gradient was used: 0→1 min, 25% B; 1→6 min, 25%→98% B; 6→6.5 min, 98% B; 6.5→7 min, 98%→25% B; 7→10 min, 25% B.

## Protein expression and purification

The N-terminal domain of Gal-8 (UniProt entry O00214 isoform 1), consisting of residues 4-158 was produced by Lund Protein Production Platform. Luria-Bertani (LB) medium supplemented with 50 μg ml$^{-1}$ kanamycin was inoculated to OD$_{600}$ = 0.1 with an overnight culture of *E. coli* TUNER(DE3) / pET26b(+)_hGal8N_4-158-M. The culture was grown at 25 °C with shaking at 250 rpm. At OD$_{600}$ = 0.46, the temperature was

lowered to 18 °C. At OD$_{600}$ = 0.9, IPTG (isopropyl β-D-1-thiogalactopyranoside) was added to 1 mM. Twenty hours after induction, the cells were harvested by centrifuging at 8000 × *g*, 4 °C, for 20 min, and the pellets were stored at −80 °C. The pellets were resuspended in MEPBS buffer (140 mM NaCl, 2.7 mM KCl, 10 mM NaH$_2$PO$_4$, 1.8 mM KH$_2$PO$_4$, pH 7.4, 2 mM EDTA, 4 mM 2-mercaptoethanol) supplemented with three tablets of Complete Protease Inhibitor, EDTA-free (Roche) and a pinch of DNase I. The cell suspension was passed two times through a French Pressure Cell at 18,000 psi. The lysate was ultracentrifuged in a Ti 50.2 rotor, 45,000 rpm, 60 min, 4 °C. The supernatant was passed through a Minisart NML Plus 0.45 μm syringe filter (Sartorius) and used for affinity chromatography.

A 19 ml lactosyl Sepharose column was connected to an ÄKTA Avant chromatography system (GE Healthcare). The run was performed at room temperature while fractions were collected at 6 °C. The flow rate was 2 mL min$^{-1}$. The column was equilibrated with 5 CV MEPBS and the sample was applied. The column was then washed with 10 CV MEPBS, and bound protein was eluted with 5 CV MEPBS + 150 mM lactose. Fractions containing Gal-8N were pooled and concentrated using Pierce concentrator spin columns with molecular weight cutoff (MWCO) 10 kDa (Thermo Scientific) at 4 °C. The sample was transferred to dialysis tubing (Spectra/por, 6-8000 MWCO) and dialyzed for ~18 h against PBS (140 mM NaCl, 2.7 mM KCl, 10 mM NaH$_2$PO$_4$, 1.8 mM KH$_2$PO$_4$, pH 7.4, 2 mM EDTA) with several changes of buffer. After concentration, the protein was at 13.1 mg ml$^{-1}$ in PBS. The yield was estimated at 220 mg Gal-8N per litre of *E. coli* culture and the purity is >95% as estimated by SDS-PAGE (sodium dodecyl sulphate–polyacrylamide gel electrophoresis). Aliquots were flash-frozen in liquid nitrogen and stored at -80 °C.

## Protein crystallization

Compound **11** was dissolved to 20 mM in 100% dimethyl sulphoxide (DMSO). This was mixed with 10 mg ml$^{-1}$ Gal-8N at a protein:ligand ratio of 19:1 to obtain a final ligand concentration of 1 mM and 5% DMSO. Initial crystallisation conditions were identified using the Morpheus screen (Molecular Dimensions).

Crystals were grown using the sitting drop vapour diffusion method in 24-well NeXtal crystallisation plates (Qiagen). Drops consisted of 2 μl Gal-8N at 13.1 mg ml$^{-1}$ in PBS buffer mixed with 2 μl of reservoir solution consisting of 35% (w/v) Morpheus precipitant mix 4, 0.16 M Morpheus alcohol mix, 61 mM MES, 39 mM imidazole, pH 7.5. Morpheus precipitant mix 4 is made up of 25% (w/v) each of methane pentane diol (MPD), polyethylene glycol (PEG) 1000 and PEG 3350. Morpheus alcohol mix contains 0.2 M each of 1,6-hexanediol, 1-butanol, 1,2-propanediol, 2-propanol, 1,4-butanediol and 1,3-propanediol (1.2 M total). Crystals were harvested from the drops in LithoLoops (Molecular Dimensions) and flash-frozen in liquid nitrogen without additional cryoprotection.

## X-ray data collection and structure determination

Data were collected at the BioMAX beamline of the MAX IV synchrotron, Lund, Sweden[37] at 100 K and a wavelength of 0.8000 Å. The beamline was equipped with a DECTRIS Eiger 16 M detector. The number of images collected was 3600, each with a rotation range of 0.1°. The best crystal diffracted to 1.08 Å. The data were processed using the EDNA_proc pipeline[38] which includes the software XDS[39] and Aimless[40]. Data quality statistics are presented in Supplementary Table 1. There are two independent copies of Gal-8N in the asymmetric unit.

The structure was determined by refining the coordinates of the protein only from an unpublished Gal-8N complex in the same space group, obtained by molecular replacement using PDB entry 5GZC as search model, against the new dataset. Difference electron density maps clearly showed compound **11** bound to both Gal-8N molecules in the asymmetric unit. Coordinates and a geometry description for compound **11** were generated using the Grade2 server version 1.3.2 (Global Phasing Ltd) using the SMILES string as input. Model building was carried out in Coot[41] and the structure was refined using phenix.refine[42] including anisotropic B-factor refinement for all atoms, including water molecules.

## Competitive fluorescence polarisation experiments

Human galectins-1, -3, and -8N were expressed and purified as described previously[8,23,24]. Competitive fluorescence polarization assays were performed with a PHERAstar FS plate reader with software PHERAstar Mars version 2.10 R3 (BMG, Offenburg, Germany) and Tecan Spark multimode microplate reader (Tecan Trading AG, Switzerland), where a fluorescence anisotropy of fluorescent-tagged probes was measured by excitation at 485 nm and emission at 520 nm. Experiments were performed at 20 °C with Gal-1 at 0.5 μM, Gal-3 at 0.2 μM, and Gal-8N at 0.4 μM. Fluorescent probes used for experiments were 3,3′-dideoxy-3-[4-(fluorescein-5-yl-carbonylaminomethyl)-1$H$-1,2,3-triazol-1-yl]-3′-(3,5-di-methoxybenzamido)-1,1′-sulfanediyl-di-β-$D$-galactopyranoside at 0.02 μM for Gal-1 and -3 and 2-(fluorescein-5-yl-carbonylamino)ethyl β-$D$-galactopyranosyl(1→4)-2-acetamido-2-deoxy-β-$D$-glucopyranosyl(1→3)-β-$D$-galactopyranosyl(1→4)-β-$D$-glucopyranoside at 0.1 μM for Gal-8N in each well. Compounds were first dissolved in pure DMSO at 20 mM concentration and later diluted in 5% DMSO in PBS solution to 5-7 different concentrations. Each concentration was tested in duplicate. The highest inhibitor concentrations tested were 1.5 mM. The average values of $K_d$ and SEM (standard error mean) were calculated from 4 to 8 duplicate measurements, showing 10–90% inhibition.

## Thermodynamic properties of ligand binding to Gal-8N determined by ITC measurements

Isothermal titration calorimetry (ITC) experiments were conducted using a Microcal PEAQ-ITC isothermal titration calorimeter (Malvern Instruments, Malvern Panalytical Ltd) at 25 °C. Lyophilized Gal-8N was first dissolved in PBS buffer (with a 4.3 mM β-mercaptoethanol) with a pH = 7.4 and Gal-8N was dialyzed with a 0.5–3 mL Slide-A-Lyzer™ 7k MWCO Dialysis Cassette in the same buffer. To remove residual lactose, Gal-8N was additionally purified using Superdex 75 Increase column equilibrated in PBS with 5% DMSO. Fractions containing Gal-8N were collected and concentrated with Amicon® Ultra-0.5 Centrifugal Filter Device with a 10k MWCO to a final concentration of about 40 μM. $A_{280}$ was measured using the NanoDrop™ One Microvolume UV-VIS spectrophotometer and the concentration of Gal-8N was calculated (ε = 11460 L mol$^{-1}$ cm$^{-1}$ was used). 16 mM stock solutions of **1**, **11**, and **29** were prepared in pure DMSO. The 16 mM stock solutions were diluted to 800 μM with pure PBS and then to the final concentration of 560 μM (the final content of DMSO was 5%). Titrations were performed with 19 injections of Gal-8N inhibitors **1**, **11** (560 μM) and **29** (800 μM) with a following settings: reference power = 10.0 μcal s$^{-1}$, feedback = high, stir speed = 750, Initial delay = 100 s, first spacing = 200 s, the other spacings = 150 s.

Raw thermograms were exported and integrated with the NITPIC (version 2.0.7)[43] and then analyzed with SEDPHAT (version 15.2)[44,45]. Parameter and error analysis were performed using the Monte-Carlo approach for nonlinear regression with 1000 iterations as implemented in SEDPHAT.

## Data availability

The coordinates and structure factors for the complex of Gal-8N with compound **11** have been deposited in the Protein Data Bank with accession number 9FYJ. Supplementary information includes synthesis protocols, $^1$H, $^{13}$C, $^{19}$F NMR and HPLC chromatograms, protein purification protocol, X-ray crystallography, ITC and FP data, LE and LLE calculations, MD simulation data, crystal structure of **11** in complex with Gal-8N and LNnT fluorescent probe binding data.

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

## Acknowledgements

The financial support of this work from the European Union's Horizon2020 program under the Marie Skłodowska-Curie grant agreement No. 765581 (project PhD4GlycoDrug; www.phd4glycodrug.eu), EUTOPIA PhD Co-tutelle Programme grant, The Swedish Research Council (Grant no. 2020-03317) and the Slovenian Research Agency (Grants P1-0208 and J1-50026) is gratefully acknowledged. COST actions CA18103 (Innogly) and CA18132 (GLYCONanoPROBES) are also gratefully acknowledged. We thank Consortium for Functional Glycomics, which provided a tetrasaccharide (LNnT) with a 2-azidoethyl linker for a Gal-8N probe synthesis (The synthesis of Gal-8N probe can be found in Supplementary Information). We also thank Barbro Kahl Knutson from the Department of Laboratory Medicine, Lund University for assistance with the fluorescence polarisation experiments and Lund Protein Production Platform for Gal-8N expression and purification. We acknowledge the MAX IV Laboratory for beamtime on the BioMAX beamline under proposal 20220263. Research conducted at MAX IV, a Swedish national user facility, is supported by Vetenskapsrådet (Swedish Research Council, VR) under contract 2018-07152, Vinnova (Swedish Governmental Agency for Innovation Systems) under contract 2018-04969 and Formas under contract 2019-02496. This research was supported by the Ministry of Education, Science, and Sport (MIZŠ) of the Republic of Slovenia and the European Regional Development Fund OP20.05187 RI-SI-EATRIS.

## Author contributions

E.P., U.J.N., D.T.L. and M.A. designed the study. E.P., F. S. and M.H. performed chemical reactions and competitive fluorescence polarization measurements. H.L. performed the protein expression and purification. E.P., T.T., A.S. and M.A. prepared virtual libraries, performed molecular docking studies and molecular dynamics. A.S. performed Energy Decomposition Analysis. E.P., M.P., J.L., and M.A. designed ITC experiments. E.P. and M.P. carried out ITC experiments. J.A.F. and D.T.L. performed the crystallography and data analysis. M.A. supervised the research. E.P., U.J.N., D.T.L., H.L., A.S., T.T., M.P., J.L., and M.A. prepared the manuscript and Supplementary information.

## Competing interests

U.J.N. and H.L. are shareholders in Galecto Biotech Inc., a company developing galectin inhibitors. All other authors declare no competing interests.
