## [Transparent Peer Review file · Communications Chemistry]

Nanomolar inhibitor of the galectin-8 N-terminal domain binds via a non-canonical cation- π interaction

Corresponding Author: Professor Marko Anderluh

Version 0:

Reviewer comments:

Reviewer #1

(Remarks to the Author)

The Work entitled "Nanomolar inhibitor of the galectin-8 N-terminal domain binds via a non-canonical cation- interaction" presents the development and evaluation of a small library of galectin-8N inhibitors that exhibit high nanomolar K_d values. The thermodynamic analysis of the binding of small group of selected inhibitors reveals important differences in enthalpic and/or entropic contributions to binding. In particular, authors show that binding of 2-O-propargyl-D-galactoside was found to strongly increase the binding enthalpy suggesting that the ethynyl group can successfully replace the carboxylate group when targeting the water-exposed guanidine moiety of a critical arginine residue, a novel type of interaction.

The work is well done and nicely complements, biochemical, structural and computational methods. It is also well written and presented with a good selection of Figures and tables and their displayed information. Therefore I think the work is suitable for publication in Communications Chemistry.

However I believe some minor issues related to the underlying interpretation of the proposed novel ethynyl interaction need some further work and/or analysis.

The Key issue is as follows: The authors correctly identify that (page 9) "The most intriguing feature of 11 is its affinity for Gal-8N, as it reached high nanomolar affinity for Gal-8N (K_d value of 800 nM), which is somewhat unexpected when comparing 11 with 29 " Moreover, "The thermodynamic analysis shows that the 2-O-carboxymethylene moiety (with a free carboxylate) of 29 affects binding with a substantial increase in binding enthalpy and a slight loss of entropy compared to 1. .. Comparison of thermodynamic parameters between 11 and 1 shows an unexpected gain in enthalpic contribution with a further loss in binding entropy compared to 29, suggesting that contributions other than classical hydrophobic interactions govern the binding of 11. ". Finally they state that.. "The important question to be answered is what is the origin of the observed enthalpic gain, especially when comparing 11 with 29? "

To explain this key issue and novel aspect of the work they present several hypothesis and performed some computational analysis. They performed MD of the complexes and also performed QM based calculations and analyze the corresponding HOMO-LUMO. In my opinion the proposed hypothesis and explanations are not well supported by the data and further analysis is required to clarify this issue. Specific suggestions follow:

1) To better analyze the different and comparative thermodynamic contributions to the binding free energy of 1, 11 and 29 authors should perform more suitable calculations. For example they could perform classical TI calculations switching between 11 and 29 using for example two steps separating electrostatic and vdw contributions

2) Authors state that "that water-exposed arginines form entropy-driven interactions" and that "The reason for this could lie in active site loosely bound water molecules with high internal energy (and therefore not stable in their environment), termed "hot" water molecules. The displacement of these loosely bound water molecules by a ligand into the bulk phase with fully tetra-coordinated state would lead to enthalpy gain and entropic penalty." To analyze the potential presence and characteristics of these "hot" waters authors could analyze the Water Sites (WS) adjacent to the Arginine. There are several works and free tools showing how to characterize WS (or hydration sites). See for example <https://doi.org/10.1093/bioinformatics/btv411>.

3) If the hypothesis underlying the binding of compound 11 binds are correct, similar behaviour is expected for compound 8 since it has a phenyl group which is also expected to perform non covalente cation-pi interactions with the R45. Why was this compound not analyzed thermodynamically?

4) The authors state "studied with quantum mechanical calculations and indicate that the LUMO of Arg45 of Gal-8N (Fig. 4c, depicted in blue) points towards the HOMO electrons of propargyl alkyne (Fig. 4c, depicted in red) of compound 11, but without evident HOMO-LUMO overlap meaning that the covalent nature of this interaction is negligible. Yet, the distance between the guanidine C-atom and the terminal C-atom of the alkyne is 3,3 Å (Fig. 4c), which is in line with a distance threshold for classical cation-interactions as these occur within a distance of 6 Å²⁷. ". To really characterize the nature of this interaction authors could compute the QM interaction energy and compare it to that with other hydrophobic non charged groups such as that found in compounds 8 and 22

5) The reason why 11 binds with a higher enthalpy than 29 is counterintuitive but can be explained by fact that the ethynyl moiety has a lower desolvation enthalpy than the free carboxylate (where release of tightly coordinated waters around carboxylate can lead to a loss in enthalpy and gain in entropy), resulting in a higher overall binding enthalpy. As for point 1 to really understand the balance of enthalpy and entropy in the desolvation of these two groups authors should perform in water calssical or even QM/MM based Thermodynamic integration calculations.

Reviewer #2

(Remarks to the Author)

The manuscript "Nanomolar inhibitor of the galectin-8 N-terminal domain binds via a non-canonical cation-pi interaction" by Puric et al describes the synthesis of numerous new galectin inhibitors based on a known thiogalactoside. The authors aimed at specificity for galectin 8N, a potential therapeutic target. All molecules were evaluated in comparative FP assays against Gal-1, -3, 8C, and -8N. While selectivity could be improved, affinity remained in the nM range. The authors then characterized selected compounds by ITC and argue for a new alkyne-Arg cation interaction, which has been demonstrated by X-ray crystallography and assigned to an electrostatic rather than a covalent bond by molecular modeling.

The paper describes a new phenomenon which will be of interest to the community and might serve for the design of novel inhibitors in a wide range of proteins.

The current work is sound, written well and merits publication in Commun Chem after a few modifications detailed below.

Major points:

1. Protein interactions are a different situation, however alkyne-cation interactions are reported in synthetic chemistry, this should be cited and mentioned at least. See Nagy et al, ChemComm, 2016, 52, 2311 and Fressigne et al., J Org Chem 2023, 88, 14494
2. Description of synthesis of the LNnT FP probe is missing. Direct titrations should also be depicted.
3. For ITC only one replicate is shown, please depict all replicate titrations in the SI. This is mandatory to allow the discussion of the enthalpy and entropy differences. Numbers of replicates must be indicated in Fig 3.

minor points:

1. in the SI, dot but not comma should be used consistently in numbers.
2. Yields with 2 decimals suggest precision but the weighing error is probably higher. E.g. "62.78 % yield (105 mg)"
3. the same is probably true for 5 decimals in HRMS
4. ¹³C NMR and MS are missing for all compounds 6 and 17, 20, 23, 26
5. cmps 9, 10, 11, miss titles in the experimental section
6. for cmp 11, the following signal is described: "4.96 (dd, J = 25.9, 12.8 Hz, 2H)," since the NMRs are unfortunately not assigned, it would be interesting to understand which proton pair couples with 25.9 Hz. Especially because that same coupling constant is not found again in any other signal. When looking at the transcript, this is probably an AB system...
7. NMR spectra must be shown in a zoom that the signals are legible. Peak picking can be removed and intensity must be magnified. Solvent peaks are not too interesting.
8. page 14, line 438: pdb code should be given
9. page 14, line 426 the unit L/molcm misses a *

Reviewer #3

(Remarks to the Author)

see attached file

Reviewer #4

(Remarks to the Author)

This manuscript by Puric et al. describes the synthesis and characterization of inhibitors of galectin-8 N-terminal domain, and the authors present the cation- π interaction of an ethynyl group and arginine as a new possibility for inhibitor design. For galectin-8, which is thought to play an important role in angiogenesis and lymphangiogenesis, the authors synthesized a series of inhibitor candidates and analyzed their affinity and specificity. As a result, it was revealed that 2-O-propargyl-D-galactose with an ethynyl group has unexpected affinity for galectin-8N. Crystal structure analysis and QM calculations suggest that the cation- π interaction between the ethynyl group and arginine is responsible for this affinity. This article is technically sound and the discussion is approximately adequate, giving new information into this research field. However, in order to improve the quality of the article, we suggest that the following several points be considered.

Major points:

- 1) As the authors argue, non-canonical cation- π interaction is promising for novel ligand design. However, the affinity and selectivity of compound 29 are better than those of 11. In Figure 3, the value of the enthalpy gain seems to be comparable between compounds 11 and 29. From these results, the authors should discuss the binding ability and selectivity of compound 29. The X-ray structures of galectin-8N/29 complex is very interesting and may be helpful for this discussion. The authors should attempt to X-ray structure determinations of compounds 29, 16a, 16b, and 16c with high affinities to galectin-8N. How did the authors generate a starting model for MD simulation of 29 (Lines 250-253)?
- 2) Line154-157: Compounds 16a, 16b, and 16c have also high affinities to galectin-8N, but the authors did not discuss them due to low ligand efficiency and ligand-lipophilicity. Possible interactions of these compounds with proteins should be discussed.
- 3) Lines 168-170 and 236-242: Different results appeared for galectin-8 and galectin-3 with respect to the enthalpy gain and entropy penalty. One possible reason is simply stated in Results (Line 170), but only a different reason is stated in Discussion (Line 237). More specific explanation about the former reason would be helpful for the readers.
- 4) Line 250: Arg45 and Arg69 are not seen in Supplementary figure 102, while these residues interact with the ligand in the docking model of Figure 2f. Which model is more correct?
- 5) Line393: An unpublished coordinate of the same space group is used for the structure refinement. It would be better to describe how the authors obtained the unpublished coordinate (e.g., molecular replacement).

Minor point:

- 6) Line77: 14-fold -> 6.8-fold
- 7) Line157: Supplementary Table 3 -> Supplementary Table 4
- 8) Line184: Legend of Fig. 4, wrong color description.
- 9) Line336: Solvent D -> Solvent B?
- 10) Line438: 8*** -> 9FYJ
- 11) Supplementary information: An overall structure illustration of galectin-8N/11 complex is helpful to understand the manuscript for readers in other research fields.

Reviewer #5

(Remarks to the Author)

I co-reviewed this manuscript with one of the reviewers who provided the listed reports. This is part of the Communications Chemistry initiative to facilitate training in peer review and to provide appropriate recognition for Early Career Researchers who co-review manuscripts.

Version 1:

Reviewer comments:

Reviewer #1

(Remarks to the Author)

The Work revised version of the work entitled "Nanomolar inhibitor of the galectin-8 N-terminal domain binds via a non-canonical cation- π interaction" presents an improved version of the original manuscript, where the authors made their best efforts to address my (which were minor suggestions), as well as the other, reviewer's concerns.

Therefore I believe in its present form the work is suitable for publication.

Specific comments follow:

- 1) My first and main concern was related to the unexpected binding behavior of compound 11 with Gal-8N, marked by a high affinity and unique thermodynamic properties and the authors proposed hypotheses and use computational methods (MD simulations and QM calculations) to explain these findings, which my feeling was required additional evidence.

In the revised version the authors did a thorough bonding analysis of 11 and 29. First, preliminary additional analysis of MD simulations was performed on Gal-8N by using affinity clustering

(see article DOI: 10.1126/science.1136800) To dissect the specific binding contributions of compound 11, also a quantum mechanics Energy Decomposition Analysis (EDA) was performed.

The results show that the acetylene and Arg45 fragments have a π - π^* interaction and the

predicted bonding energy is -3.4 kcal/mol., which further supports the findings. The work was done by Dr. Anders Sundin

from Lund University and was added as a co-author.

2) My second remark was related to "hot waters". Although the authors performed further analysis they were not conclusive, nonetheless they result an important addition to the work.

3) Third remark was related to the similarity of compounds 8 and 11. The authors correctly point out that compounds 8a-f had higher Kd values and therefore a lower affinity for Gal-8 than propargyl-substituted compound 11. Therefore, they considered it unnecessary to analyse further.

Remarks 4 and 5 have been correctly addresses but not pursued further due to the length of the manuscript.

Reviewer #2

(Remarks to the Author)

Thank you very much for sending this revised version of an interesting paper. In this revision, the authors have satisfactorily addressed all my previous requests and comments.

Reviewer #3

(Remarks to the Author)

I thank the authors for addressing the points raised in the referee report. The manuscript has been improved and is ready to be published. Congratulations on a nice piece of work. I recommend the paper for publication.

Reviewer #4

(Remarks to the Author)

The authors have taken notice of our comments on the previous version. However, we would like to suggest a minor point.

Point-by-point response to the reviewers' comments

Reviewer #4 (Remarks to the Author):

Remark 5) Line393: An unpublished coordinate of the same space group is used for the structure refinement. It would be better to describe how the authors obtained the unpublished coordinate (e.g., molecular replacement).

Reply 4) Derek, please add the reply.

Perhaps Dr. Derek T. Logan might forget to respond (Line 452-453 in the new version). We expect the following corrections, for example, "an unpublished Gal-8N complex in the same space group, which was solved by molecular replacement using a model derived from PDB ID: XXX (or AlphaFold2 [ref])".

After this minor correction, the manuscript will be acceptable for the publication in Communications Chemistry.

Reviewer #5

(Remarks to the Author)

I co-reviewed this manuscript with one of the reviewers who provided the listed reports. This is part of the Communications Chemistry initiative to facilitate training in peer review and to provide appropriate recognition for Early Career Researchers who co-review manuscripts.

Point-by-point response to the reviewers' comments

Journal: Communications Chemistry, Manuscript ID: COMMSCHEM-24-0507-T

Title: "Nanomolar inhibitor of the galectin-8 N-terminal domain binds via a non-canonical cation- π interaction"

Author(s): Edvin Purić, Mujtaba Hassan, Fredrik Sjövall, Tihomir Tomašič, Mojca Pevec, Jurij Lah, Jaume Adrover Forteza, Anders Sundin, Hakon Leffler, Ulf J. Nilsson, Derek T. Logan and Marko Anderluh

Reviewer #1 (Remarks to the Author):

The Work entitled “Nanomolar inhibitor of the galectin-8 N-terminal domain binds via a non-canonical cation- π interaction” presents the development and evaluation of a small library of galectin-8N inhibitors that exhibit high nanomolar K_d values. The thermodynamic analysis of the binding of small group of selected inhibitors reveals important differences in enthalpic and/or entropic contributions to binding. In particular, authors show that binding of 2-O-propargyl-D-galactoside was found to strongly increase the binding enthalpy suggesting that the ethynyl group can successfully replace the carboxylate group when targeting the water-exposed guanidine moiety of a critical arginine residue, a novel type of interaction.

The work is well done and nicely complements, biochemical, structural and computational methods. It is also well written and presented with a good selection of Figures and tables and their displayed information. Therefore, I think the work is suitable for publication in Communications Chemistry.

However, I believe some minor issues related to the underlying interpretation of the proposed novel ethynyl interaction need some further work and/or analysis.

The Key issue is as follows: The authors correctly identify that (page 9) “ The most intriguing feature of **11** is its affinity for Gal-8N, as it reached high nanomolar affinity for Gal-8N (K_d value of 800 nM), which is somewhat unexpected when comparing **11** with **29** “ Moreover, “ The thermodynamic analysis shows that the 2-O-carboxymethylene moiety (with a free carboxylate) of **29** affects binding with a substantial increase in binding enthalpy and a slight loss of entropy compared to **11**. .. Comparison of thermodynamic parameters between **11** and **1** shows an unexpected gain in enthalpic contribution with a further loss in binding entropy compared to **29**, suggesting that contributions other than classical hydrophobic interactions govern the binding of **11**. “. Finally they state that.. “The important question to be answered is what is the origin of the observed enthalpic gain, especially when comparing **11** with **29**? “

To explain this key issue and novel aspect of the work they present several hypotheses and performed some computational analysis. They performed MD of the complexes and also performed QM based calculations and analyze the corresponding HOMO-LUMO. In my opinion the proposed hypothesis and explanations are not well supported by the data and further analysis is required to clarify this issue. Specific suggestions follow:

Remark 1) To better analyze the different and comparative thermodynamic contributions to the binding free energy of **1**, **11** and **29** authors should perform more suitable calculations. For example, they could perform classical TI calculations switching between **11** and **29** using for example two steps separating electrostatic and vdw contributions.

*Reply 1) To answer the question, we did a thorough bonding analysis of **11** and **29**. First, additional analysis of MD simulations was performed on Gal-8N by using affinity clustering (see article DOI: 10.1126/science.1136800) with bound compounds **11** and **29**, both structures based on the crystal of compound **11** in complex with Gal-8N (please see the new Supplementary figure 107). The clusters of the compound **11** trajectory did not feature contact between Arg45 and the acetylene, which is present in the crystal. This is probably caused by the lack of parameters for general π - π orbital interactions in molecular mechanics force fields. The clusters of both compound **11** and of compound **29** had one thing in common, the methylene group shielded the hydrophobic face of the guanidine moiety of Arg45 from contact with water. To dissect the specific binding contributions of compound **11**, a quantum mechanics Energy Decomposition Analysis (EDA) was performed. The conclusion of this analysis is next:*

molecular Orbital Energy-Level Diagram obtained by the EDA analysis indicates that the acetylene and Arg45 fragments have a π - π^* interaction and the predicted bonding energy is -3.4 kcal/mol. A specific dissection of energy bonding contributions is given as a separate supplementary file, while the new Figure 5 and the following text was added to the manuscript: “Furthermore, to dissect specific binding energy contributions of compound **11** in complex with Gal-8N, a quantum mechanics Energy Decomposition Analysis (EDA) was performed²⁵. Such an analysis requires that the structure is split into two interacting fragments. Since the compound **11** has multiple interactions with the protein and we wanted to investigate the interaction between Arg45 and the acetylene, the ligand had to be modified. The first fragment was chosen to be the galactose C2 substituent bearing the acetylene, and second fragment was Arg45 together with any parts of the protein or ligand that may influence the electronic structure of Arg45 (Fig. 5a). That is, the stacking Arg69 with its water mediated salt bridge to Glu89 and the hydrogen bond network from Arg45 through galactose 4-OH to His65 to Asp49 were included. The Molecular Orbital Energy-Level Diagram obtained by the EDA analysis (Fig. 5b) indicates that the acetylene and Arg45 fragments have a π - π^* interaction and the predicted bonding energy is -3.4 kcal/mol. Plotting the bonding molecular orbital indicates a π - π^* interaction between the acetylene and the guanidinium ion (Supplementary figure 109). It has previously been shown that the LUMO of the guanidinium ion is π^* ²⁶. The observation was corroborated by a Natural Orbitals for Chemical Valence (NOCV) deformation density plot that describes the charge flow between acetylene and Arg45 when the two fragments are combined. Electrons flow from red to blue, i.e. from acetylene towards the guanidinium group thus showing the origin of π - π^* interaction (Fig. 5c). A specific dissection of energy bonding contributions is given as a separate supplementary file.”

Additional references were also cited: 25 and 26 in the revised manuscript.

We believe that such analysis answers to the Reviewers suggestion to separate binding contributions to electrostatic and vdw contributions (EDA does even go further) and validates our presumption that π - π interactions are formed between an acetylene of **11** and guanidine of Arg45. This work was done by Dr. Anders Sundin from Lund University and was added as a co-author.

Remark 2) Authors state that “that water-exposed arginines form entropy-driven interactions” and that “The reason for this could lie in active site loosely bound water molecules with high internal energy (and therefore not stable in their environment), termed “hot” water molecules. The displacement of these loosely bound water molecules by a ligand into the bulk phase with fully tetra-coordinated state would lead to enthalpy gain and entropic penalty.” To analyze the potential presence and characteristics of these “hot” waters authors could analyze the Water Sites (WS) adjacent to the Arginine. There are several works and free tools showing how to characterize WS (or hydration sites). See for example <https://doi.org/10.1093/bioinformatics/btv411>.

Reply 2) We thank the Reviewer for a very useful suggestion. Analysis of MD trajectories with WATCLUST did not identify any water sites in proximity to Arg45. However, we did not want to go deep into the analysis of the water sites, as it is obvious that these molecules should be replaced with either a carboxylate (of **29**) or an alkyne residue (of **11**). The important question that we wanted to answer is how the binding of both groups differentiate in a thermodynamic sense. An in-depth analysis with WATCLUST combined with Autodock could give us some useful data, but we believe that the results of a scoring function can vary substantially depending on the docking software used. This is why we opted to perform ITC experiments rather than relying on calculations and to try to interpret thermodynamic data.

Remark 3) If the hypothesis underlying the binding of compound **11** binds are correct, similar behaviour is expected for compound **8** since it has a phenyl group which is also expected to perform non covalente cation- π interactions with the R45. Why was this compound not analyzed thermodynamically?

Reply 3) Compounds 8a-f had higher K_d values and therefore a lower affinity for Gal-8 than propargyl-substituted compound 11. In fact, these compounds had similar K_d values (or even higher) to the unsubstituted derivative (1) as the difference was not significant at best. Thus, even if a cation- π interaction is formed between a (substituted) phenyl residue and the guanidine of Arg45, it is most probably overcompensated by high desolvation penalties, proving that no favourable binding has taken place. We considered it unnecessary to analyse compounds that do not provide any overall affinity gain over unsubstituted derivative 1.

Remark 4) The authors state “studied with quantum mechanical calculations and indicate that the LUMO of Arg45 of Gal-8N (Fig. 4c, depicted in blue) points towards the HOMO electrons of propargyl alkyne (Fig. 4c, depicted in red) of compound **11**, but without evident HOMO-LUMO overlap meaning that the covalent nature of this interaction is negligible. Yet, the distance between the guanidine C-atom and the terminal C-atom of the alkyne is 3,3 Å (Fig. 4c), which is in line with a distance threshold for classical cation-interactions as these occur within a distance of 6 Å²⁷. “. To really characterize the nature of this interaction authors could compute the QM interaction energy and compare it to that with other hydrophobic non charged groups such as that found in compounds 8 and 22

Reply 4) The reviewer is right. We could compute the QM interaction energy and compare it to that of other hydrophobic, uncharged groups, such as those found in compounds 8 and 22. For example, we can hypothesize that 22 can form van der Waals interactions or ion-induced dipole interactions with Arg45. To prove this, we should probably perform additional ITC measurements and compare the experimental data with the calculated data. This would expand the already rather lengthy communication-type manuscript into the full-length manuscript. We deliberately opted for a communication-type manuscript in order to strengthen the focus on cation-alkyne(π) interactions. Also, we have obtained only one co-crystal structure (11), while we should take docked poses of compounds 8 and 22 for further calculations. Since we cannot exclude the error in the predicted/docked pose, we fear that these results would not provide unequivocal evidence for the nature of the interactions involved between 11 and Gal-8.

Remark 5) The reason why **11** binds with a higher enthalpy than **29** is counterintuitive but can be explained by fact that the ethynyl moiety has a lower desolvation enthalpy than the free carboxylate (where release of tightly coordinated waters around carboxylate can lead to a loss in enthalpy and gain in entropy), resulting in a higher overall binding enthalpy.

As for point 1 to really understand the balance of enthalpy and entropy in the desolvation of these two groups authors should perform in water classical or even QM/MM based Thermodynamic integration calculations.

Reply 5) The reviewer is right, but for the sake of manuscript length (as stated in the Reply 4), we did not want to expand the manuscript, but rather offer a possible explanation for the observed phenomenon. Another issue may be found in a ligand interaction diagram for 29 after affinity clustering (please see new Supplementary figure 108). Compound 29 was found to have the carboxylate stationary in all of the major clusters and analysis of the trajectory featured a single water mediated contact with Arg45 during 36% of the trajectory, but the carboxylate likely also has multiple interactions to both Arg45 and to Arg69 mediated with two water molecules. This means that there is a high probability that the carboxylate remains partially solvated and that these water molecules are not replaced upon binding, but rather, they form

water-bridged contacts with arginines. This is why just QM/MM calculations may not be enough to depict the whole enthalpy-entropy issue and a more dynamic approach is needed.

Reviewer #2 (Remarks to the Author):

The manuscript "Nanomolar inhibitor of the galectin-8 N-terminal domain binds via a non-canonical cation- π interaction" by Puric et al describes the synthesis of numerous new galectin inhibitors based on a known thiogalactoside. The authors aimed at specificity for galectin 8N, a potential therapeutic target. All molecules were evaluated in comparative FP assays against Gal-1, -3, 8C, and -8N. While selectivity could be improved, affinity remained in the nM range. The authors then characterized selected compounds by ITC and argue for a new alkyne-Arg cation interaction, which has been demonstrated by X-ray crystallography and assigned to an electrostatic rather than a covalent bond by molecular modeling.

The paper describes a new phenomenon which will be of interest to the community and might serve for the design of novel inhibitors in a wide range of proteins.

The current work is sound, written well and merits publication in Commun Chem after a few modifications detailed below.

Major points:

Remark 1) Protein interactions are a different situation, however alkyne-cation interactions are reported in synthetic chemistry, this should be cited and mentioned at least. See Nagy et al, ChemComm, 2016, 52, 2311 and Fressigne et al., J Org Chem 2023, 88, 14494

Reply 1) We agree with the Reviewer#2 and we thank him/her for a very useful tip. We have mentioned this on the Page 10 of the revised manuscript in the next sentence "Alkyne-cation interactions have been reported several times in synthetic chemistry, but to our knowledge no such interactions have been found between a ligand and a protein, which is why we have labelled them "non-canonical cation- π interactions".

Remark 2) Description of synthesis of the LNnT FP probe is missing. Direct titrations should also be depicted.

Reply 2) The Consortium for Functional Glycomics provided a tetrasaccharide for the synthesis of Gal-8N probe shown below.

Figure. The structure of LNnT probe coupled with 5-carboxyfluorescein. The probe is generally used in a competitive fluorescence polarisation assay when determining binding affinities of galectin-8N inhibitors.

To clearly describe the synthesis of the LNnT FP probe we have added the following text on pages 32-33 of the Supplementary Information: "The Consortium for Functional Glycomics provided the tetrasaccharide (LNnT) for the synthesis of the Gal-8N probe. LNnT with 2-azidoethyl linker was converted to free amine in a Staudinger reaction and later coupled with 5-carboxyfluorescein. This probe is generally used in a competitive fluorescence polarisation assay to determine the binding affinities for Gal-8N inhibitors (Please see <https://doi.org/10.1039/C8OB01354C> and <https://doi.org/10.1016/j.ejmech.2021.113664>). The structure and synthesis of the LNnT probe are shown below (Supplementary scheme 9)."

The *Reaction scheme 1* can be found in the *Supplementary Information*, referred as *Supplementary Scheme 9*.

Reaction scheme 1. Reagents and conditions: a) H_2 , Pd/C, MeOH, rt, overnight; b) 5-carboxyfluorescein, BOP, HOBT, DIPEA, DMSO, rt, overnight.

We have added the direct titration in the *Supplementary Information*, as suggested by the *Reviewer #2* (please see new *Supplementary figure 111*).

Figure 2. Binding of galectin-8, galectin-8N and galectin-8C to LNnT fluorescent probe.

Remark 3) For ITC only one replicate is shown, please depict all replicate titrations in the SI. This is mandatory to allow the discussion of the enthalpy and entropy differences. Numbers of replicates must be indicated in Fig 3.

Reply 3) We have included all replicate titrations in the *Supplementary Information* (*Supplementary Figure 97*).

Minor points:

1. in the SI, dot but not comma should be used consistently in numbers.

We have changed this throughout the text.

2. Yields with 2 decimals suggest precision but the weighing error is probably higher. E.g. "62.78 % yield (105 mg)".

Yields are now provided without decimals.

3. the same is probably true for 5 decimals in HRMS.

HRMS data is now provided with 4 decimals.

4. ¹³C NMR and MS are missing for all compounds 6 and 17, 20, 23, 26.

¹³C and HRMS data are provided only for final compounds that were used in a competitive fluorescence polarisation assay. ¹H NMR reports for intermediates can be found in Supplementary Information (subsection Synthesis).

5. cmps 9, 10, 11, miss titles in the experimental section.

We thank the reviewer for this observation. Titles are added.

6. for cmp 11, the following signal is described: "4.96 (dd, J = 25.9, 12.8 Hz, 2H)," since the NMRs are unfortunately not assigned, it would be interesting to understand which proton pair couples with 25.9 Hz. Especially because that same coupling constant is not found again in any other signal. When looking at the transcript, this is probably an AB system...

The authors thank the reviewer for this observation. The signal at 4.96 ppm is an AB system or AB quartet (ABq), which is now clearly defined.

7. NMR spectra must be shown in a zoom that the signals are legible. Peak picking can be removed and intensity must be magnified. Solvent peaks are not too interesting.

We have zoomed NMR spectra as requested.

8. page 14, line 438: pdb code should be given.

PDB code is given.

9. page 14, line 426 the unit L/molcm misses a *.

We have corrected the mistake.

Reviewer#3

Remark 1) The work described in the manuscript deals with the modification of the C-2 position in D-galactoside and its impact to binding affinity and selectivity to Gal-8N. As far as known, the C-2 modification is the least considered for affinity and selectivity tuning and it is more used for tuning of drug-like properties of inhibitors. Therefore, this is a novel and interesting aspect. On the other hand, as follows from the results in Table 1, C-2 modification does not influence binding and selectivity significantly. The best inhibitors, although structurally very different, exhibit a 2-fold (inhibitor **11**), 3-fold (inhibitor **16b**), or max 4-fold (inhibitors **16a** and **29**) increase in affinity to Gal-8N compared to the unmodified parent compound **1**. Selectivity to other galectins was improved a maximum of 2 times in the whole synthesized series. This advocates that C-1 and C-3 modifications are still the way to get galectin inhibitors with notably improved affinity and selectivity. This should be included in the discussion.

Reply 1) We thank the reviewer for recognizing the novelty of C2-modifications. As requested, we have included the following lines into discussion (Page 9 of the revised manuscript): »Despite these improvements, we must emphasize that the selectivity over other galectins in the entire series was improved by a maximum factor of 2. This indicates that modifications of positions 1 and 3 of the D-galactose core are still the likely option for the development of galectin inhibitors with significantly improved affinity and selectivity, and the overall affinity and selectivity is the consequence of a finely tuned combination of substitutions at positions 1, 2 and 3«.

Remark 2) Is it possible that substitution at C-2 affect somehow the orientation of substituents at C-1 and C-3 (and therefore binding affinity of the molecule)? What is the conformational behavior of free (unbound) inhibitors? It would be useful to have a comparison with Supplementary Figures 100 and 103 which are for bound ligands.

*Reply 2) We thank the Reviewer#3 for an interesting observation and question. To answer the question, we have performed the MD simulation of **1**, **11** and **29** in water only (unbound ligands), and have added this simulation in the SI (Supplementary figures 104-106). The comparison of MD simulation data of **1** with **11** and **29** show that the torsion profiles of bonds of substituents at positions 1 and 3 of the D-galactose core remain almost the same irrespectively of the substituent at position 2. This shows that substituents at position 2 of **11** and **29** leave flexibility of substituents at positions 1 and 3 almost unaffected.*

*To highlight this in the manuscript, we have added the following text on page 10: "...but it could also be the result of steric hindrance of the substituent at position 2, which could affect the flexibility of the substituents at positions 1 and 3. To exclude this possibility, we performed the MD simulation of **1**, **11** and **29** in water only (Fig. S104-S106). Comparison of the MD simulation data of **1** with **11** and **29** shows that the bond torsional profiles of the substituents at positions 1 and 3 remain almost the same, regardless of the substituent at position 2 indicating that the substituents at position 2 of **11** and **29** leave the flexibility of the substituents at positions 1 and 3 almost unaffected."*

Remark 3) The most interesting part of the work is the comparison of the binding for **11** and **29** and deciphering the interaction that drives higher binding for propargyl derivative **11**. A substantial part of the discussion is devoted to the explanation of enthalpic and entropic contributions to the binding based on ITC results. This part needs, in my opinion, a bit more work. The enthalpy contribution can be influenced by the buffer used in the experiment (<https://doi.org/10.1016/bs.mie.2015.08.025>). In order to be sure about enthalpy (and therefore

entropy as well) contribution, authors are asked to measure ITC data at least in two buffer systems for studied compounds **1**, **11**, and **29**.

Reply 3) We fully agree with the reviewer that the buffer used can significantly influence the measured enthalpy changes. The reason lies in the possible (de)protonation of the molecules involved in the binding, which is coupled to the (de)protonation of the buffer component. To check whether (de)protonation occurs during the observed binding, we followed the binding of compound **1** with galectin-8N with ITC in TRIS buffer, which has an ionization enthalpy of about 50 kJ/mol (for PBS this value is zero; J Biol Chem. 1956, 218, 961). If (de)protonation were to occur during binding, the binding enthalpy determined in the TRIS buffer would differ by at least 50 kJ/mol from that determined in PBS. Our measurements and analysis show that the values are within the experimental error (Supplementary Table 3). From this we conclude that there are no (de)protonation effects on binding. Since compounds **11** and **29** are direct derivatives of **1**, no (de)protonation is expected upon binding to galectin-8N as all compounds have carboxylates in an ionized form at certain pH value fixed by the corresponding buffer (PBS). This is why we believe that additional ITC experiments with **11** and **29** in TRIS are not necessary.

Table 3. Comparison of standard thermodynamic parameters of **1** in PBS and TRIS buffer.

Buffer	K_d [μ M]	ΔH [kJ/mol]	$-T\Delta S$ [kJ/mol]	ΔG [kJ/mol]
TRIS	4.5	-50	19	-30.7
PBS	5	-44	14	-30.2

Other comments/suggestions:

Remark 4) The title is probably too optimistic – I found inhibitors with sub-micromolar affinities in the manuscript. “Nanomolar” in the title evokes much more potent compounds. Why do authors use “non-canonical” cation- π interaction for the interaction of cation with alkyne? This is well known type of noncovalent interaction.

Reply 4) Indeed, our molecules have sub-micromolar affinity at best, but still, this is the high-nanomolar concentration range. Especially in the case of lectins where we cope with rather open, flat and polar binding sites, reaching sub-micromolar affinity is a substantial achievement with just a few successful examples in the literature. We agree that the improvement over **1** was not huge, yet with our compounds we did cross the magical “nanomolar threshold” and the term was not used to falsely impress the readers, but rather to distinguish these new molecules from our (and others) previous work. If the Reviewer#3 insists, we will remove “nanomolar” from the title.

We agree that cation-alkyne reactions are already known in synthetic chemistry (we have added two new references to support this), but to our knowledge this is the first report of a cation-alkyne(π) interaction between a protein and a ligand (we have performed a search of all alkyne-containing ligands through the PDB and found no such example) and as such may have an important impact on medicinal chemistry. To corroborate the title, we have added the following text in the discussion part explaining why we have labelled this interaction as “non-canonical”: “Alkyne-cation interactions have been reported several times in synthetic chemistry, but to our knowledge no such interactions have been found between a ligand and a protein, which is why we have labelled them “non-canonical cation- π interactions”.”

Remark 5) In the introduction – I did not find any sub-nM inhibitor in reference 2.

Reply 5) We thank the reviewer for this observation. There are only low nM and micromolar galectin-3 inhibitors in the reference 2, so we have modified the text accordingly.

Remark 6) Introduction, paragraph 3: “...and inhibition one of the two CRDs should be enough.” Enough for what? Probably for the inhibition of whole tandem-repeat galectin.

Reply 6) We agree with the reviewer and have added this explanation.

Remark 7) Introduction, paragraph 3: “Recently, it was discovered that these could be replaced by an α -linked halogenated benzyl...” – should be probably “ α -linked halogenated phenylsulfanyl”

Reply 7) We agree with the reviewer and have changed the sentence.

Remark 8) Results, paragraph 3: “...while it exhibited much better ligand efficiency and ligand lipophilicity efficiency compared to the triazoles **16a-c**, as shown in Supplementary Table 3.” –should be Supplementary Table 4.

Reply 8) We thank the reviewer for this observation and we have changed it.

Remark 9) Discussion, paragraph 2: ...2-O-propargyl moiety (**11**) is slightly less flexible in the bound state than the 2-O-carboxymethylene (**29**) (Fig. S100 and Fig. S103). The latter could explain why **29** has a lower entropic penalty than **11** upon binding.” – I do not understand this conclusion, because increased flexibility will increase entropy.

*Reply 9) We agree with the Reviewer#3 as it should be the other way around. In fact, we wanted to say that the flexibility around O-C bond at position 2 of the D-galactoside that defines flexibility of the residue is more flexible in **11** than in **29** (light orange diagram in Fig. S100 and Fig. S103). We have changed this sentence accordingly: “...-O-propargyl moiety (**11**) is slightly more flexible in the bound state than the 2-O-carboxymethylene (**29**) (Fig. S100 and Fig. S103).”.*

Remark 10) In Supplementary Information – new compounds are poorly characterized – for many of them ^{13}C NMR is missing.

Reply 10) In the Supplementary Information we attached ^1H , ^{13}C , ^{19}F (when necessary) and HPLC chromatograms of all final compounds that were tested in a competitive fluorescence polarisation assay. Please see pages S34-S85. HRMS data of all final compounds can be found in the Supplementary Information (subsection Synthesis). This is done for all end compounds, as is the case with medicinal chemistry-focused papers.

Based on what I've written, I recommend the manuscript for publication after major revisions. I would be happy to evaluate the revised version.

Reviewer #4 (Remarks to the Author):

This manuscript by Puric et al. describes the synthesis and characterization of inhibitors of galectin-8 N-terminal domain, and the authors present the cation- π interaction of an ethynyl group and arginine as a new possibility for inhibitor design. For galectin-8, which is thought to play an important role in angiogenesis and lymphangiogenesis, the authors synthesized a series of inhibitor candidates and analyzed their affinity and specificity. As a result, it was revealed that 2-O-propargyl-D-galactose with an ethynyl group has unexpected affinity for galectin-8N. Crystal structure analysis and QM calculations suggest that the cation- π interaction between the ethynyl group and arginine is responsible for this affinity. This article is technically sound and the discussion is approximately adequate, giving new information into this research field. However, in order to improve the quality of the article, we suggest that the following several points be considered.

Major points:

Remark 1) As the authors argue, non-canonical cation- π interaction is promising for novel ligand design. However, the affinity and selectivity of compound **29** are better than those of **11**. In Figure 3, the value of the enthalpy gain seems to be comparable between compounds **11** and **29**. From these results, the authors should discuss the binding ability and selectivity of compound **29**. The X-ray structures of galectin-8N/**29** complex is very interesting and may be helpful for this discussion. The authors should attempt to X-ray structure determinations of compounds **29**, **16a**, **16b**, and **16c** with high affinities to galectin-8N. How did the authors generate a starting model for MD simulation of **29** (Lines 250-253)?

*Reply 1) We agree that the most potent and selective compound in the series is **29** and not **11**. However, the increase in affinity of **29** is not surprising, while it was not evident for **11**, and even more puzzling was the result from ITC where **11** had the most negative enthalpy. This is why we have co-crystallized **11** first and presented it in this communication-type manuscript. Also, **11** has a potential advantage over **29** as has only one free carboxylate, while **29** has two. This means that **11** might passively permeate cells to target intracellular Gal-8, while **29** is most probably too polar. Co-crystallization of other ligands will be attempted in the following work, but we aim for even higher affinity ligands for co-crystallization studies. The starting model for MD simulation of **29** was the docked pose of **29** with Gal-8, as stated on the page 11: “Gal-8N in docking complex with inhibitor **29** or crystal structure of **11** in complex with Gal-8N (PDB entry: 9FYJ) was used as an input for molecular dynamics simulation using Desmond³⁰.”.*

Remark 2) Line154-157: Compounds **16a**, **16b**, and **16c** have also high affinities to galectin-8N, but the authors did not discuss them due to low ligand efficiency and ligand-lipophilicity. Possible interactions of these compounds with proteins should be discussed.

*Reply 2) We thank Reviewer#4 for this comment. A discussion on the possible interactions of **16a-c** can be found in lines 222-225 of the original submission. We have also added a new text in the line 126, where we thought it would make most sense: “The carboxylate-triazoles **16a-c** were designed to possess both a phenyl ring and a free carboxylate since a phenyl moiety can offer cation- π interactions while a carboxylate can be involved in ionic interactions or salt bridges with Arg45 and/or Arg69.”.*

Remark 3) Lines 168-170 and 236-242: Different results appeared for galectin-8 and galectin-3 with respect to the enthalpy gain and entropy penalty. One possible reason is simply stated

in Results (Line 170), but only a different reason is stated in Discussion (Line 237). More specific explanation about the former reason would be helpful for the readers.

Reply 3) We thank the reviewer for this intriguing comment. Indeed, in our previous publication shows that addition of a sulphate at position 2 is beneficial due to entropic gain: "The somewhat higher affinities of the 2-O-sulfated 19 and 20, as compared to the non-sulfated 15 and 16, were also confirmed by ITC, with the effect being slightly greater for the fluorinated ligand, as expected. The affinity increases upon sulfation (15 vs. 19 and 16 vs. 20, respectively) is driven by entropy alone, whereas the change in binding enthalpy is unfavourable. These changes in binding enthalpy and entropy induced by the sulfate group, viz. $\Delta\Delta H > 0$ and $-T\Delta\Delta S < 0$, are expected because the attractive Coulomb interaction between opposite charges (i.e., the guanidine and sulfate groups) is entirely due to favourable entropy and involves unfavourable enthalpy." Although in the previous work we have proven that ionic interaction between guanidine and a sulphate at position 2 of the D-galactose is entropy-driven, a direct comparison is not simple since in the current work we do not have a sulphate. This is why we have modified the sentence in line 236 to "This was not expected, as we found in our previous work that water-exposed arginines form entropy-driven ionic (Coulomb) interactions, yet a direct comparison with the data presented in this work is not trivial since in the previous work the ionic interaction was involving the guanidine and sulphate groups and the latter (sulphate groups) are not present in our current set of molecules".

Remark 4) Line 250: Arg45 and Arg69 are not seen in Supplementary figure 102, while these residues interact with the ligand in the docking model of Figure 2f. Which model is more correct?

Reply 4) This is indeed true and is the consequence of different amino acid numbering in the non-optimized crystal structure, and can be seen also in the figure 99. We have modified both figures as requested so that the correct numbering is used.

Remark 5) Line393: An unpublished coordinate of the same space group is used for the structure refinement. It would be better to describe how the authors obtained the unpublished coordinate (e.g., molecular replacement).

Reply 4) Derek, please add the reply.

Minor point:

6) Line77: 14-fold -> 6.8-fold.

We have corrected the mistake.

7) Line157: Supplementary Table 3 -> Supplementary Table 4.

We have corrected the mistake.

8) Line184: Legend of Fig. 4, wrong color description.

We have corrected the mistake.

9) Line336: Solvent D -> Solvent B?

We have corrected the mistake.

10) Line438: 8*** -> 9FYJ.

We have corrected the mistake.

11) Supplementary information: An overall structure illustration of galectin-8N/11 complex is helpful to understand the manuscript for readers in other research fields.

We thank the Reviewer #4 for this suggestion and have included the structure illustration of galectin-8N/11 complex in the Supplementary Information (please see **Supplementary figure 110**).

Supplementary figure 110. 3D structure of galectin-8N in complex with **11** (PDB ID: 9FYJ): grey cartoon representation of galectin-8N, **11** in sticks: carbon in green, oxygen in red, sulphur in yellow, chlorine in forest green, nitrogen in blue). The distance between the terminal propargyl carbon atom of **11** and the terminal Arg45 nitrogen atom of galectin-8N is labelled.

Reviewer #5 (Remarks to the Author):

I co-reviewed this manuscript with one of the reviewers who provided the listed reports. This is part of the Communications Chemistry initiative to facilitate training in peer review and to provide appropriate recognition for Early Career Researchers who co-review manuscripts.

Point-by-point response to the reviewers' comments

Journal: Communications Chemistry, Manuscript ID: COMMSCHEM-24-0507-T

Title: "Nanomolar inhibitor of the galectin-8 N-terminal domain binds via a non-canonical cation- π interaction"

Author(s): Edvin Purić, Mujtaba Hassan, Fredrik Sjövall, Tihomir Tomašič, Mojca Pevec, Jurij Lah, Jaume Adrover Forteza, Anders Sundin, Hakon Leffler, Ulf J. Nilsson, Derek T. Logan and Marko Anderluh

2nd Review

Reviewer #4 (Remarks to the Author):

Point-by-point response to the reviewers' comments

Reviewer #4 (Remarks to the Author):

Remark 5) Line393: An unpublished coordinate of the same space group is used for the structure refinement. It would be better to describe how the authors obtained the unpublished coordinate (e.g., molecular replacement).

Reply 4) Derek, please add the reply.

Reply 1) Indeed, we forgot to answer this comment. As requested, we have changed the sentence:

“The structure was determined by refining the coordinates of the protein only from an unpublished Gal-8N complex in the same space group against the new dataset.”

with the following sentence:

*“The structure was determined by refining the coordinates of the protein only from an unpublished Gal-8N complex in the same space group, **obtained by molecular replacement using PDB entry 5GZC as search model**, against the new dataset.”*

Reviewer opinion on manuscript# COMMSCHEM-24-0507-T by Marko Anderluh:

The work described in the manuscript deals with the modification of the C-2 position in D-galactoside and its impact to binding affinity and selectivity to Gal-8N. As far as known, the C-2 modification is the least considered for affinity and selectivity tuning and it is more used for tuning of drug-like properties of inhibitors. Therefore, this is a novel and interesting aspect. On the other hand, as follows from the results in Table 1, C-2 modification does not influence binding and selectivity significantly. The best inhibitors, although structurally very different, exhibit a 2-fold (inhibitor **11**), 3-fold (inhibitor **16b**), or max 4-fold (inhibitors **16a** and **29**) increase in affinity to Gal-8N compared to the unmodified parent compound **1**. Selectivity to other galectins was improved a maximum of 2 times in the whole synthesized series. This advocates that C-1 and C-3 modifications are still the way to get galectin inhibitors with notably improved affinity and selectivity. This should be included in the discussion.

Is it possible that substitution at C-2 affect somehow the orientation of substituents at C-1 and C-3 (and therefore binding affinity of the molecule)? What is the conformational behavior of free (unbound) inhibitors? It would be useful to have a comparison with Supplementary Figures 100 and 103 which are for bound ligands.

The most interesting part of the work is the comparison of the binding for **11** and **29** and deciphering the interaction that drives higher binding for propargyl derivative **11**. A substantial part of the discussion is devoted to the explanation of enthalpic and entropic contributions to the binding based on ITC results. This part needs, in my opinion, a bit more work. The enthalpy contribution can be influenced by the buffer used in the experiment (<https://doi.org/10.1016/bs.mie.2015.08.025>). In order to be sure about enthalpy (and therefore entropy as well) contribution, authors are asked to measure ITC data at least in two buffer systems for studied compounds **1**, **11**, and **29**.

Other comments/suggestions:

1. The title is probably too optimistic – I found inhibitors with sub-micromolar affinities in the manuscript. “Nanomolar” in the title evokes much more potent compounds. Why do authors use “non-canonical” cation- π interaction for the interaction of cation with alkyne? This is well-known type of noncovalent interaction.
2. In the introduction – I did not find any sub-nM inhibitor in reference 2.
3. Introduction, paragraph 3: “...and inhibition one of the two CRDs should be enough.” Enough for what? Probably for the inhibition of whole tandem-repeat galectin.
4. Introduction, paragraph 3: “Recently, it was discovered that these could be replaced by an α -linked halogenated benzyl...” – should be probably “ α -linked halogenated phenylsulfanyl”
5. Results, paragraph 3: “...while it exhibited much better ligand efficiency and ligand-lipophilicity efficiency compared to the triazoles **16a-c**, as shown in Supplementary Table 3.” – should be Supplementary Table 4
6. Discussion, paragraph 2: ...2-O-propargyl moiety (**11**) is slightly less flexible in the bound state than the 2-O-carboxymethylene (**29**) (Fig. S100 and Fig. S103). The latter could explain why **29** has a lower entropic penalty than **11** upon binding.” – I do not understand this conclusion, because increased flexibility will increase entropy.
7. In Supplementary Information – new compounds are poorly characterized – for many of them ^{13}C NMR is missing.

Based on what I've written, I recommend the manuscript for publication after major revisions. I would be happy to evaluate the revised version.